# Impact response of lightweight steel foam concrete composite slabs: Experimental, numerical and analytical studies

**Linghui Meng**[1], **Lei Wang**[2], **Jinbo Chen**[2], **Qiang Xu**[1]*, **Bowen Liu**[1], **Minghao Yang**[1], **Shuwang Yang**[1], **Zhenhui Zhang**[1]

**1** School of Architecture & Civil Engineering, Liaocheng University, Liaocheng, China, **2** China Railway 17th Bureau Group 1st Engineering Co., Ltd, Qingdao, China

* xuqianglcu@163.com

## Abstract

This paper presents a study on the low-velocity impact response of lightweight steel foam concrete (LSFC) composite slabs. The LSFC composite slab consisted of a W-shaped steel plate, foam concrete and oriented strand board (OSB). Low-velocity impact tests on the LSFC composite slabs were conducted by employing an ultra-high heavy-duty drop hammer testing machine. The tests revealed the failure mode, impact force and displacement response of LSFC composite slabs. The effects of density and thickness of foam concrete and drop height on the peak impact force and energy absorption ratio were investigated. A finite element (FE) model was set up to predict the impact resistance of the LSFC composite slabs, and a good agreement between simulation and test results was achieved. In addition, an equivalent-single-degree-of-freedom (ESDOF) model was set up to predict the displacement response of the LSFC composite slabs under impact loading.

## 1. Introduction

At present, steel plate concrete composite slabs occupy an important position in the construction industry. However, with the change of social demand, high-rise and ultra-high-rise buildings are gradually gaining development. Currently, the main material for most composite floor slabs is concrete. However, a major problem is the self-weight, long construction period and high cost of concrete [1–3]. For high-rise buildings, due to its length and slenderness ratio is higher than the general buildings, the self-weight is also much larger than the general buildings, resulting in the load transmitted to the foundation is also increased, due to the foundation to withstand the upper part of the larger load, so that the high-rise buildings to resist seismic forces, the wind's ability to greatly reduced. Conventional steel-concrete composite slabs as the main component of the building structure, its self-weight accounted for nearly 40% of the average weight of the entire building structure [4]. If can reduce the self-weight of the slabs, it will be effective in controlling the self-weight of the entire building structure. As a new type of composite member, lightweight steel foam concrete composite slab combines the advantages of cold-formed thin-walled steel and foam concrete, integrating lightweight, high-

**Data Availability Statement:** All relevant data are within the manuscript and its Supporting Information files.

**Funding:** The author(s) received no specific funding for this work.

**Competing interests:** The authors have declared that no competing interests exist.

strength, environmental protection, energy saving, and high efficiency, which can greatly improve the performance of building slabs.

However, in recent years, there has been a gradual increase in the number of incidents in which high-rise and ultra-high-rise buildings are subjected to incidental load impacts. Under these incidental loads, the structure may undergo localized damage and cause impact collapse of the structures, which manifests itself in the form of localized or whole floor slab collapse, acting in the form of impact on the lower floor slabs and causing their destruction [5]. Contemporary studies mainly focus on the impact behavior of steel-concrete-steel (SCS) composite slabs [6–10], while the impact behavior of LSFC composite slabs is rarely studied.

Lu et al. [11] investigated the impact response of flat steel plate-concrete-corrugated steel plate (FS-C-CS) sandwich panels using drop hammer impact tests and finite element simulations and found that the bending resistance and stiffness along the span direction of the corrugated steel plate were improved compared to conventional flat panels, resulting in a significant improvement in the impact resistance of FS-C-CS panels. Cheng et al. [12] conducted low-velocity impact tests and finite element simulations on U-shaped corrugated sandwich panels to investigate the sandwich panels' dynamic response and damage state. Wang et al. [13] proposed a new stiffened rib double steel plate concrete composite panel and found that all specimens exhibited a combined failure mode of overall buckling and local indentation by drop hammer impact tests. Wang et al. [14] conducted impact tests on steel-polyurethane foam-steel-concrete-steel (SPUFSCS) panels and found that most of the impact energy was dissipated by the "soft layer" (top plate and polyurethane foam). In addition, an equivalent two degrees of freedom (TDOF) analytical model was proposed to predict the displacement response of SPUFSCS panels under impact load. An et al. [15] studied the low-speed impact response of the double-steel plate composite wall under axial load conditions through experiments and numerical calculations, found that the axial load can effectively improve the impact resistance of the component, and proposed an empirical formula to predict the maximum mid-span displacement of the double-steel plate composite wall under the impact load. Guo, Sohel, and Liu et al. demonstrated the excellent accuracy of the equivalent-single-degree-of-freedom (ESDOF) model in predicting the dynamic response of sandwich panels [16–18].

As a lightweight material, foam concrete has properties of light, sound insulation, thermal insulation [19,20], and fire resistance [21,22]. In past studies, the main focus was on foam concrete's density and compressive strength [23–26]. Reflections on the dynamic response of foam concrete still need to be improved. Due to the advantages of low density and large porosity, foam concrete has a good cushioning effect and energy absorption for impact loads [27]. Zhou et al. [28] combined foam concrete and honey-comb structure to study the dynamic response characteristics of the composite material. Guo et al. [29] and Su et al. [30] used the density and thickness of foam concrete as variables, respectively, to study its effect on the cushioning performance of foam concrete.

As a new type of prefabricated component, LSFC composite slab overcomes the limitations of the traditional SCS composite structure, which is self-heavy, complicated in process, and poor in environmental protection and energy absorption. It not only reduces the use of traditional concrete and resource consumption, but also has good energy-saving efficiency and construction speed, provides a guarantee for the safety and sustainability of the building, and has significant advantages in promoting the development of green buildings, bringing strong comprehensive benefits. In this paper, the dynamic response of LSFC composite slabs under impact loading is investigated through drop-weight impact tests and finite element (FE) simulations. In addition, an equivalent-single-degree-of-freedom (ESDOF) model was established to calculate the displacement-time histories of LSFC composite slabs. The reliability of the analysis method was verified by comparing it with the test results, which provided an

evaluation method and design basis for engineering construction. By conducting drop-weight impact tests on LSFC composite slabs. Study the impact energy, foam concrete thickness and density on the dynamic response of LSFC composite slab. Provide design parameters and theoretical basis for the construction of high-rise buildings. Then through the theoretical analysis of the method, accurately meet the slab impact resistance index. To ensure the stability and safety of the building structure under various impact loading, and to improve the survivability of the building in catastrophic events.

## 2. Experimental study

### 2.1. Specimen design

As shown in Fig 1, the LSFC composite slab is a sandwich structure. The top layer, core material and bottom layer are OSB, foam concrete and W-shaped steel plate, respectively. To investigate the effect of different parameters on the impact response of LSFC composite slabs, seven specimens were designed for the drop hammer impact test. The detailed geometric parameters of all the specimens are given in Table 1. All specimens were of the same length and width. The thickness of the W-shaped steel plate is 2 mm and the thickness of the OSB is 9 mm. The bonding between W-shaped steel plate and foam concrete relies on grip force, and the bonding between OSB and foam concrete is carried out by nail-free adhesive, produced by Glei-how company. The mixing proportions of cement, water and blowing agent were obtained from tests, as shown in Table 2. The cement type used in the test was P·O42.5 and the blowing agent was FB-602 plant-based cement blowing agent. When preparing the specimens, the molds were first supported around the W-shaped steel plates, the pouring heights were measured and marked inside the molds, and then the prepared foam concrete was poured into the molds. After 28 days of curing, the surface of the foam concrete was evenly coated with nail-free adhesive and the OSB was laid. The specimens can be tested 24 hours later.

### 2.2. Material properties

The same batch of steel plates and foam concrete were taken, and they were subjected to the metal tensile test [31], and axial compressive strength test of foam concrete [32], and the stress-strain curves of the steel plate and foam concrete are shown in Fig 2. The material property of the OSB is referred to as the material property test done by Chen et al. [33]. The material properties of the steel plate, foam concrete, and OSB are given in Table 3.

### 2.3. Test setup

The test setup used for the test was the ultra-high heavy-duty drop hammer testing machine, and the test setup is shown in Fig 3. The specimen was supported by only two support beams with a clear span of 2680 mm. The pressure beams were placed over the ends of the specimen to prevent the specimen from uplifting during rebound. Each support beam was connected to the rigid support through a ball hinge to form a simply supported boundary condition, and the drop hammer moves freely downward along two smooth guide rails. In this test, the diameter of the cylindrical hammer is 220 mm, and the hammer mass is all set to 230 kg. Three displacement sensors were used to measure the displacement of the LSFC composite slab. The measurement points of the specimen included the center point of the W-shaped steel plate (sensor A in Fig 4), the mid-span edge point of the W-shaped steel plate (sensor B in Fig 4) and the 1/4 span point of the W-shaped steel plate (sensor C in Fig 4). In addition, a high-speed camera was used to record the whole impact process. The sampling frequency of the displacement and force sensors is 200 kHz, and the camera frame rate is 2500 fps.

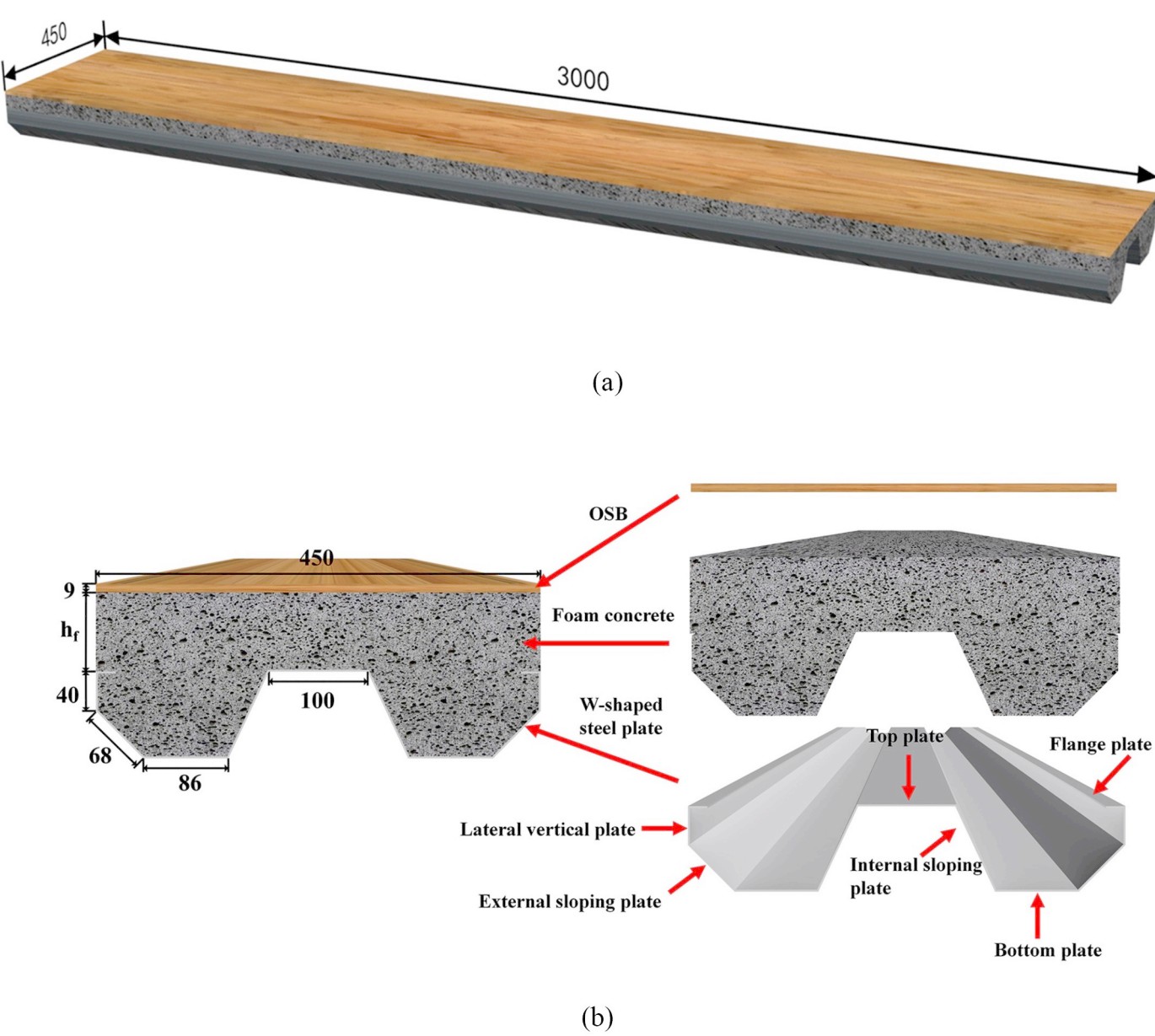

**Fig 1. LSFC composite slab (Unit: mm).** (a) Diagram of LSFC composite slab; (b) Schematic cross-section.

## 3. Test results and discussions

### 3.1. Damage mode

Fig 5A and 5B show the damage modes of LSFC composite slabs, including both overall bending and local indentation. Under the impact loading, the LSFC composite slabs produced large bending moments, resulting in vertical and diagonal cracks in the foam concrete, and the closer to the mid-span position of the LSFC composite slabs, the more concentrated the cracks were. Due to the large vertical deformation, the foam concrete near the mid-span simultaneously showed different degrees of fragments, and different degrees of interlayer debonding occurred between OSB and foam concrete. From Fig 5A, it is further observed that the impact of the hammer on the surface of OSB formed a local indentation, and the diameter of the

**Table 1. Geometric parameters of specimens.**

| Specimen | $h_f$ (mm) | $h_L$ (mm) | $\rho$ (kg/m$^3$) | $\rho_t$ (kg/m$^3$) | Basic conditions |
|---|---|---|---|---|---|
| L-H40-#1 | 40 | 2400 | 450 | 485 | The hammer mass is 230 kg and the impact position is mid-span. |
| L-H40-#2 | | | 750 | 734 | |
| L-H40-#3 | | | 600 | 622 | |
| L-H20-#1 | 20 | | | 625 | |
| L-H30-#1 | 30 | | | 620 | |
| L-H40-#4 | 40 | 2800 | | 619 | |
| L-H40-#5 | | 2000 | | 628 | |

**Note:** $h_f$ is the thickness of foam concrete on the top plate; $h_L$ is the drop height of the hammer; $\rho$ is the design value of foam concrete density; $\rho_t$ is the actual value of foam concrete density.

indentation area coincided with the diameter of the hammer, and the damage morphology of the local indentation of OSB was related to the shape of the hammer and the magnitude of the impact force. The appearance of the local indentation phenomenon further indicates that when the LSFC composite slab was subjected to external impact loading, the foam concrete and OSB were compressed and deformed, resulting in some areas of indentation. The fracture of OSB occurred around the indentation area, which was due to the stress concentration phenomenon when the edge of the hammer contacted the OSB. At the same time, due to the overall bending deformation of the LSFC composite slab, the W-shaped steel plates at the mid-span position were subjected to the extrusion force from both sides, which led to different degrees of folding deformation of the flange plates of the W-shaped steel plates. From Fig 5C, it can be seen that the LSFC composite slab also showed cracks near the support, which was caused by the support reaction force at the support.

From Fig 6A, it can be observed that there was a trace of slip between the foam concrete and the W-shaped steel plate, and the phenomenon was observed at both ends of all specimens. Bar graphs have been drawn showing the maximum value of the slip values at both ends in each specimen, as shown in Fig 6B. It is worth noting that increasing the thickness and density of the foam concrete reduced the slip values to a large extent, which is because as the thickness and density of the foam concrete were increased, the impact resistance of the LSFC composite slabs was increased and the overall deformation of the slabs gradually was decreased, which resulted in less fragmentation of the foam concrete. Thus, it was further revealed that the fragmentation of the foam concrete caused the end slip.

Fig 7 shows the impact process of L-H40-#2 recorded by the high-speed camera. With the contact of the hammer with the OSB, vertical cracks appeared in the foam concrete of L-H40-#2 at 8 ms. At 12 ms, some regions of the OSB begin to separate from the foam concrete. At 78 ms, the vertical displacement of L-H40-#2 reaches its maximum value, after which the composite slab starts to spring back.

**Table 2. The mixing proportions of foam concrete.**

| Foam concrete (kg/m$^3$) | Cement (kg/m3) | Water (kg/m3) | Foam (kg/m3) | Dilution ratio of blowing agent |
|---|---|---|---|---|
| 450 | 277.78 | 138.89 | 33.33 | 1:40 |
| 600 | 379.75 | 189.87 | 30.38 | |
| 750 | 477.71 | 238.85 | 33.44 | |

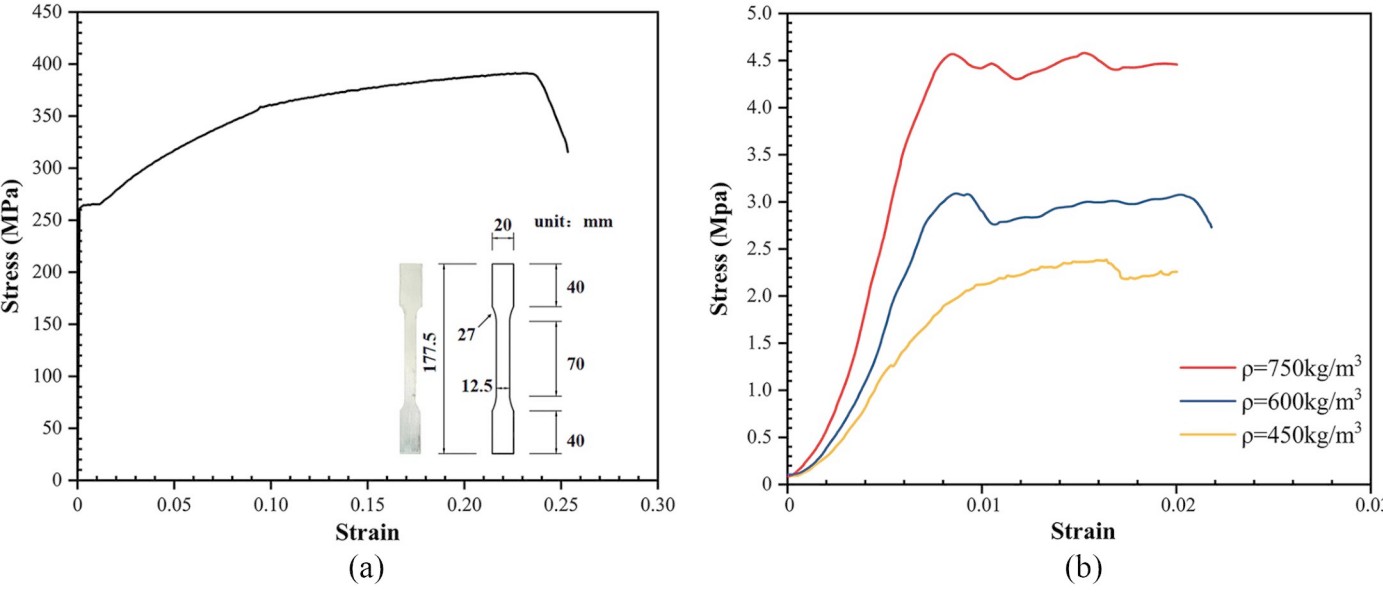

**Fig 2. Stress-strain curves.** (a) Steel plate; (b) Foam concrete.

### 3.2. Rate of dip angles change

The rate of mid-span dip angles change can more intuitively reflect the transverse deformation characteristics of W-shaped steel plates under impact load. The calculation formula for the rate of dip angles change is:

$$C = [(T_1 - T_2)/T_2] \times 100\% \tag{1}$$

where $C$ is the rate of dip angle change, $T_1$ is the dip angle value after impact, and $T_2$ is the initial dip angle value. Because foam concrete has certain elasticity and deformation ability, it can share and alleviate the impact load on the W-shaped steel plates to a certain extent, and the deformation of the W-shaped steel plates is limited. Thus, the rate of dip angles change is reduced. The effect of foam concrete on the rate of dip angle change is shown in Fig 8A and 8B. The rate of dip angle change is decreased with the increase in thickness and density of the foam concrete. In terms of thickness, as the thickness of foam concrete is increased, more tiny pores are involved in the compression deformation process, which enhances the compacting

**Table 3. Material parameters of steel plate, foam concrete and OSB.**

| Steel plate | $E_s$ (GPa) | $f_y$ (Mpa) | $f_u$ (Mpa) |
|---|---|---|---|
| — | 205.955 | 261.430 | 390.890 |
| **Foam concrete** | $E_c$ (Gpa) | $f_{cp}$ (MPa) | — |
| $\rho = 450$ | 0.29 | 2.39 | — |
| $\rho = 600$ | 0.35 | 3.09 | — |
| $\rho = 750$ | 0.50 | 4.58 | — |
| **OSB** | $E_o$ (Gpa) | $f_o$ (MPa) | — |
| — | 3.560 | 25.1 | — |

**Note:** $E_s$, $f_y$ and $f_u$ represent the elastic modulus of steel plate, yield strength and ultimate strength respectively; $E_c$, $f_{cp}$ represent the elastic modulus of foam concrete and axial compressive strength respectively; $E_o$, $f_o$ represent the average static bending stiffness and average elastic modulus of OSB respectively.

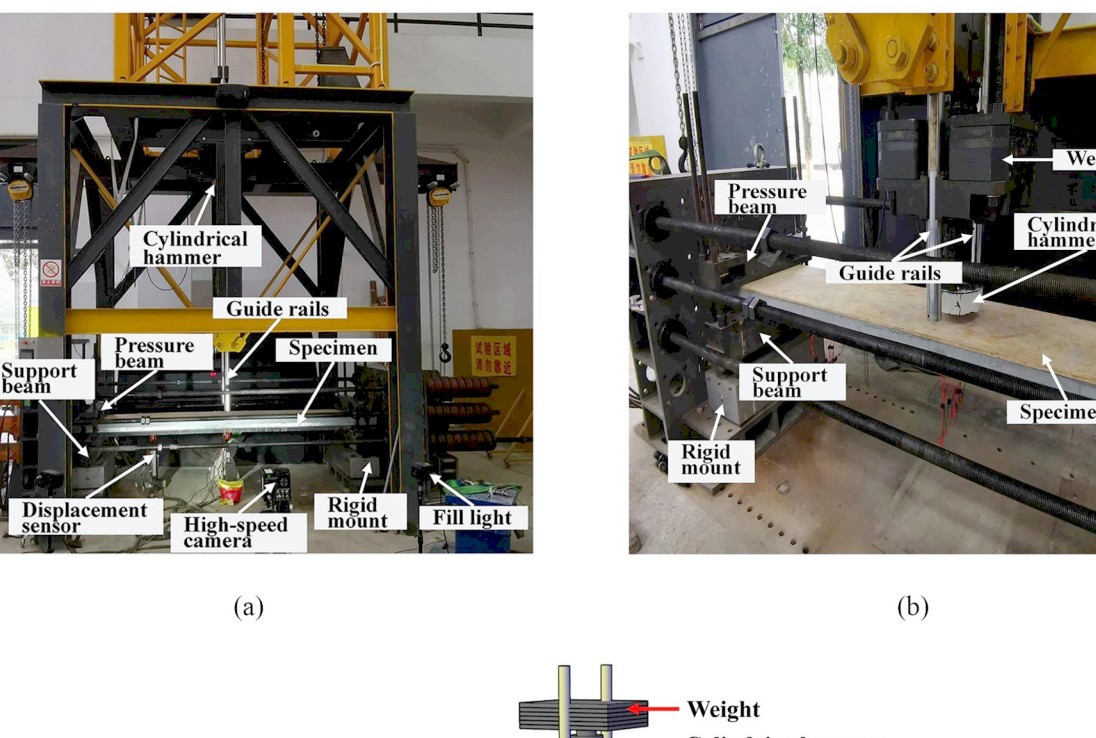

(a)                                                      (b)

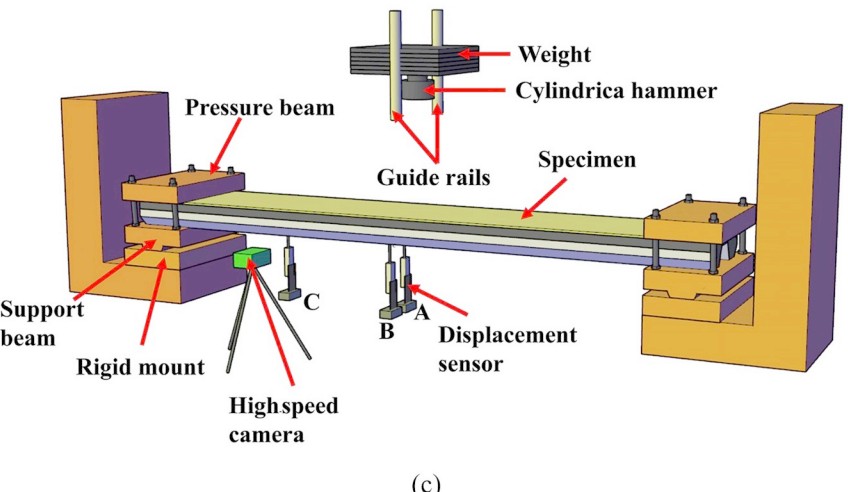

(c)

**Fig 3. Ultra-high heavy-duty drop hammer testing machine.** (a) Front view; (b) Side view; (c) 3D view.

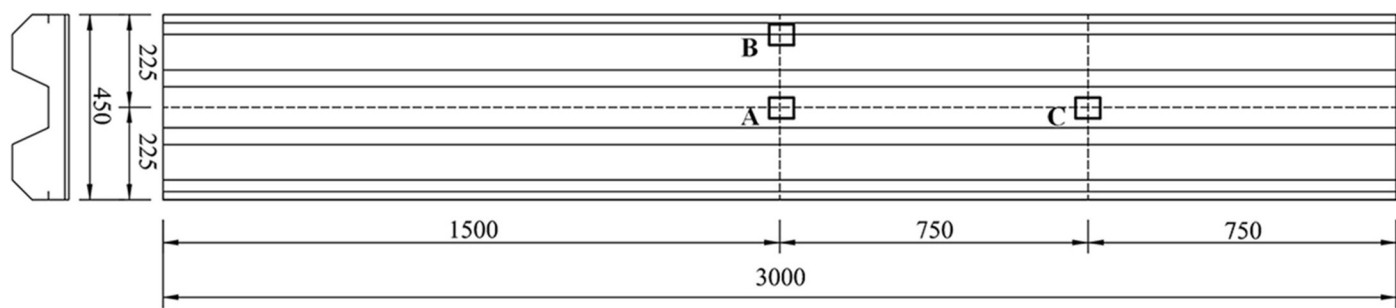

**Fig 4. Layout of displacement sensors.**

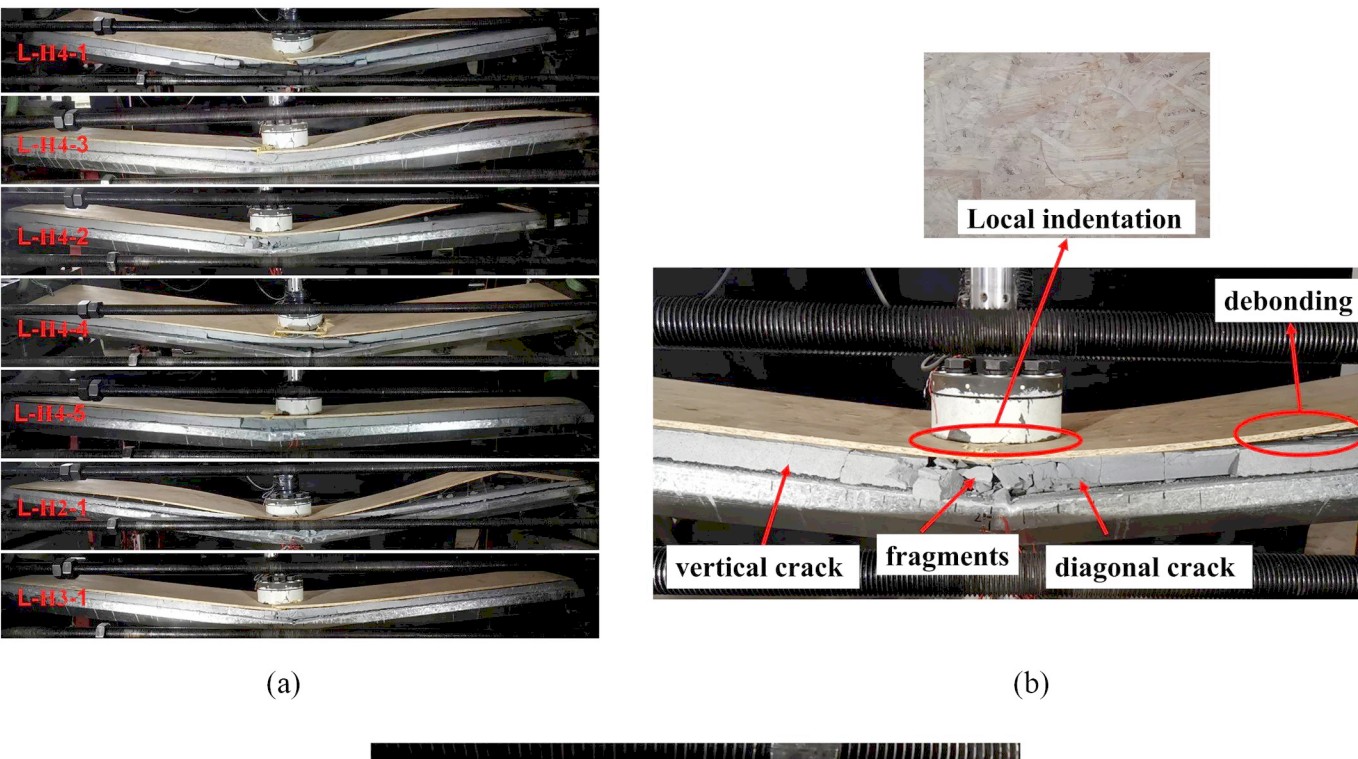

(a)                                                              (b)

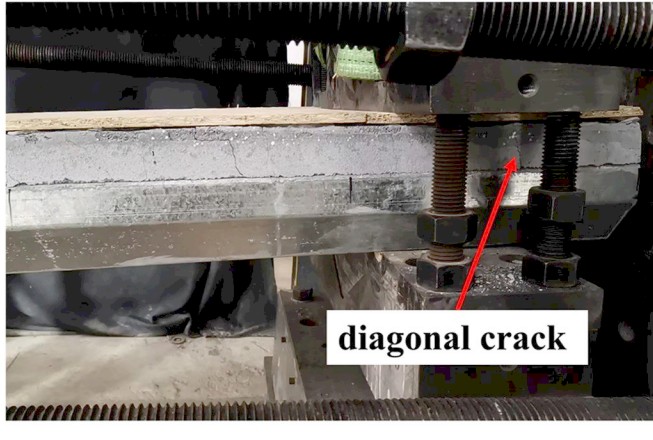

(c)

**Fig 5. Damage mode of LSFC composite slab.** (a) Overall bending; (b) Local indentation; (c) Damage at supports.

ability of foam concrete. In addition, thicker foam concrete can disperse the impact energy more widely, which reduces the damage caused by the impact. In terms of density, increasing the density of the foam concrete not only improves the stiffness and flexural strength, but also means that the internal structure is more compact, thus propagating and dispersing the impact force more effectively. From Fig 8C, it is found that the rate of dip angles change is increased as the drop height of the hammer is increased. This is attributed to the fact that the increase in drop height increases the impact energy on the W-shaped steel plate. This high-impact energy leads to an instantaneous increase in local stresses in the W-shaped steel plate, which in turn leads to a greater change in dip angle. It is worth noting that the value of Angle 4 is negative, which is due to the vertical bending deformation of the LSFC composite slab under the impact

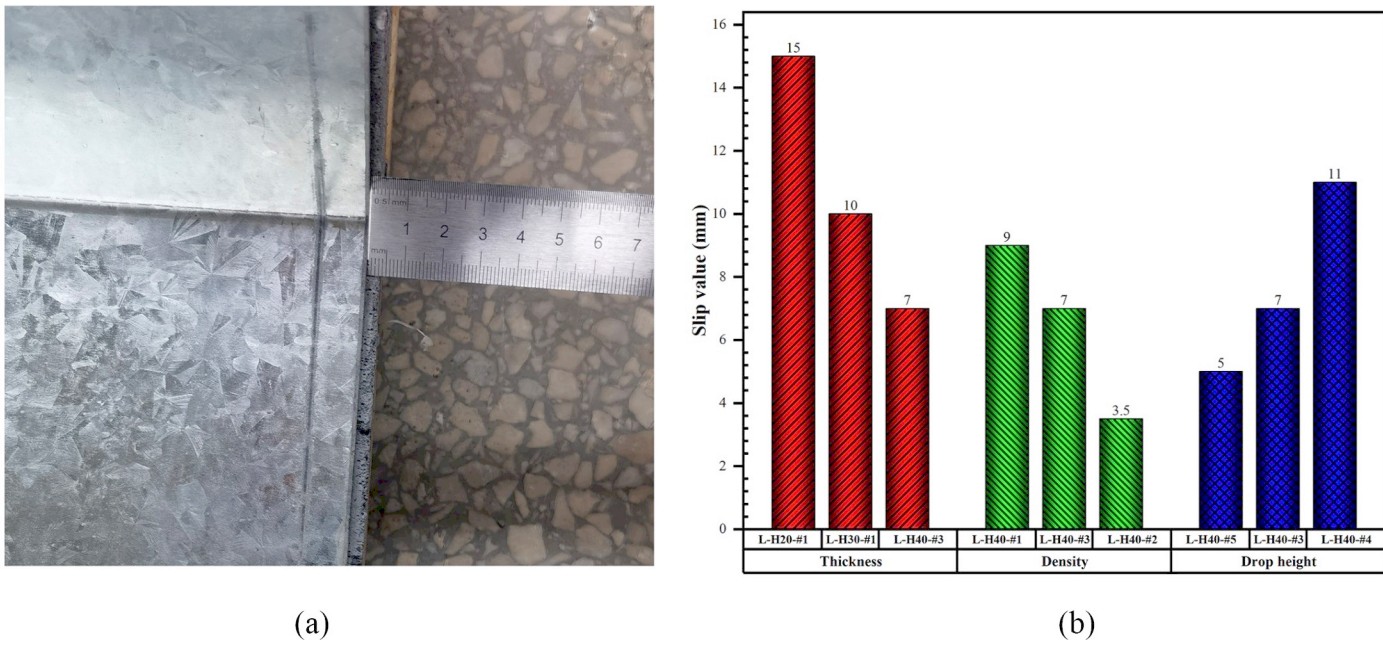

(a)                                                                                    (b)

**Fig 6. End slip diagram of LSFC composite slab.** (a) End slip diagram; (b) End slip value.

load, and the internal sloping plate and the top plate at the mid-span position are extruded, while bulging toward the outside of the W-shaped steel plate, resulting in the angle after the deformation of Angle 4 being smaller than the initial angle.

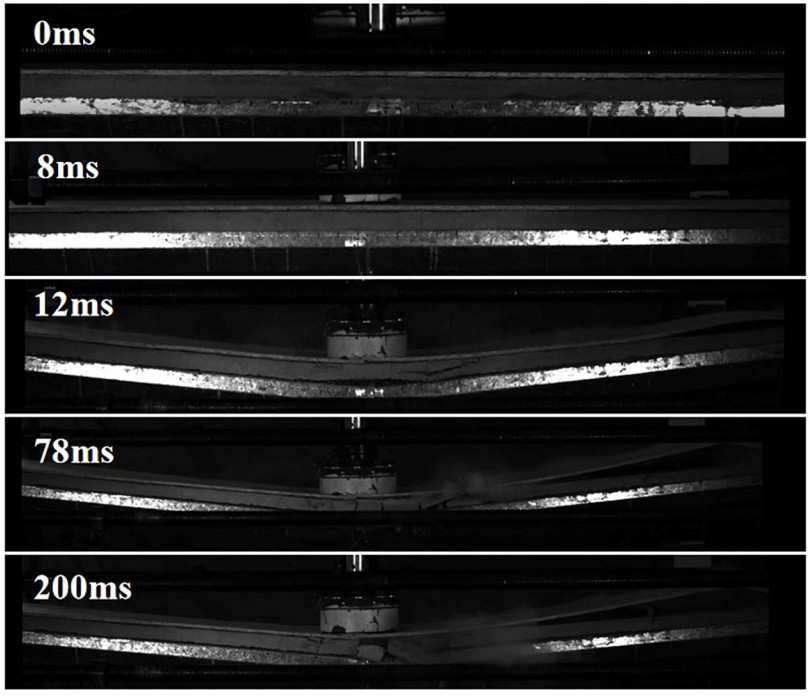

**Fig 7. Impact process of L-H40-#2.**

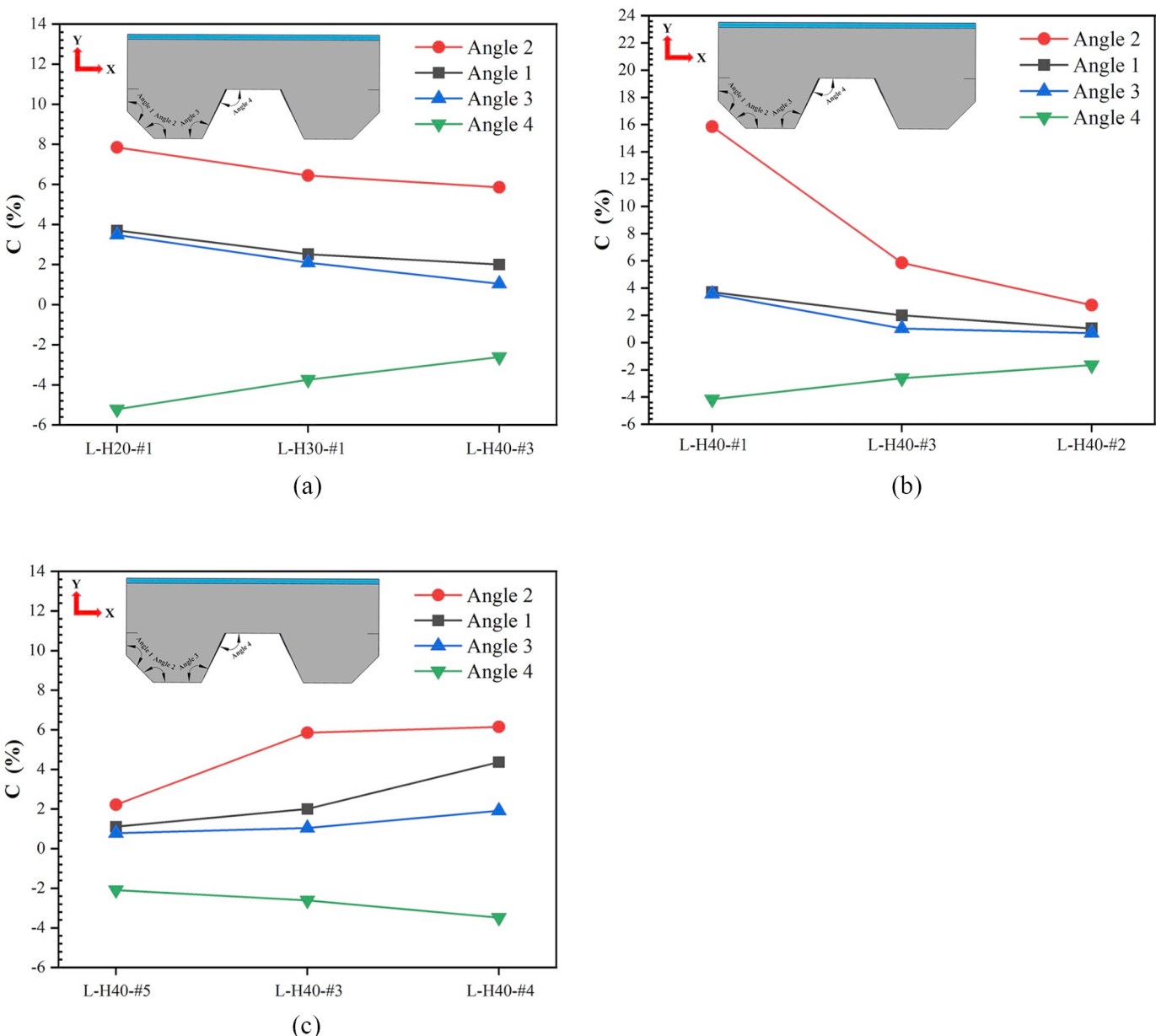

**Fig 8. Rate of change of mid-span dip angles after impact.** (a) Different foam concrete thicknesses; (b) Different foam concrete densities; (c) Different drop heights.

### 3.3. Impact force response

The typical impact force and displacement history curves of specimen L-H40-#2 are shown in Fig 9. The impact process is divided into three phases: inertial, loading, and unloading stages. The inertial stage occurs between 0–16 ms, and the impact force increases rapidly and reaches its peak. However, according to D'Alembert's principle, the change in impact force during this stage is due to inertial effects and can't represent the actual resistance of the specimen [34]. At this stage, contact is formed between the hammer and the OSB, which causes the local indentation zone of the specimen to then move downward at the same speed as the hammer, and as a result, local indentations and cracks appear on the surface of the OSB. The appearance of local

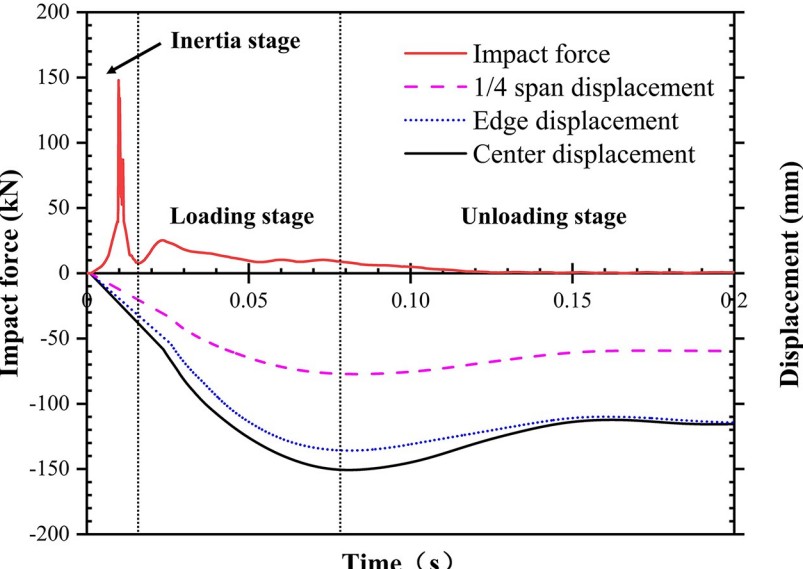

**Fig 9. Typical impact force and displacement history curves of LSFC composite slab.**

indentation also indicates that the displacement of the hammer increases faster than the displacement of the center. The loading stage occurs between 16 ms-78 ms; in this stage, the displacements at points A and B of the LSFC composite slab increase rapidly, and the displacement at point C increases more slowly relative to points A and B. The impact force slightly fluctuates in the loading stage, caused by the contact area change between the hammer and the OSB during the relative motion. At the end of the loading stage, the deformation of the LSFC composite slab reaches the maximum value. From 78 ms to 200 ms is the unloading stage, in which the impact force and displacement begin to decrease monotonically, and when the hammer is wholly separated from the slab, the impact force drops to zero, and the displacement gradually becomes constant.

The history curves of the impact force for all specimens are given in Fig 10. Table 4 lists the specific characteristic values of the impact force. According to the analysis of Fig 10 and Table 4, when the thickness of foam concrete is increased from 20 mm to 30 mm and 40 mm, $F_{max}$ $F_{max}$ is decreased by 2.10% and 25.96% respectively. This is because as the thickness of the foam concrete increases, its densification ability is strengthened, which can hinder the propagation of the impact force to a greater extent and reduce the impact force. When the density of foam concrete is increased from 450 kg/m³ to 600 kg/m³ and 750 kg/m³, $F_{max}$ is increased by 6.90% and 50.47% respectively. This is because as the density of the foam concrete increased, its internal structure became more compact and there was more resistance to the transmission of the impact force, and therefore a greater impact force was generated. When the drop height of the hammer is increased from 2.0 m to 2.4 m and 2.8 m, $F_{max}$ is increased by 6.92% and 25.06% respectively. This is because the greater the fall height, the greater the gravitational potential energy of the hammer, which is translated into greater kinetic energy and momentum during the fall.

Since the impact force remains relatively stable during the loading phase and the inertia force is negligible, the post-peak mean force can generally represent the actual resistance of the structure [35]. The post-peak mean force and peak impact force for all specimens are presented in Table 4. It can be found that for the density of foam concrete, Fmax and Fm follow the same pattern of change, that is, as the density of foam concrete is increased, both Fmax and

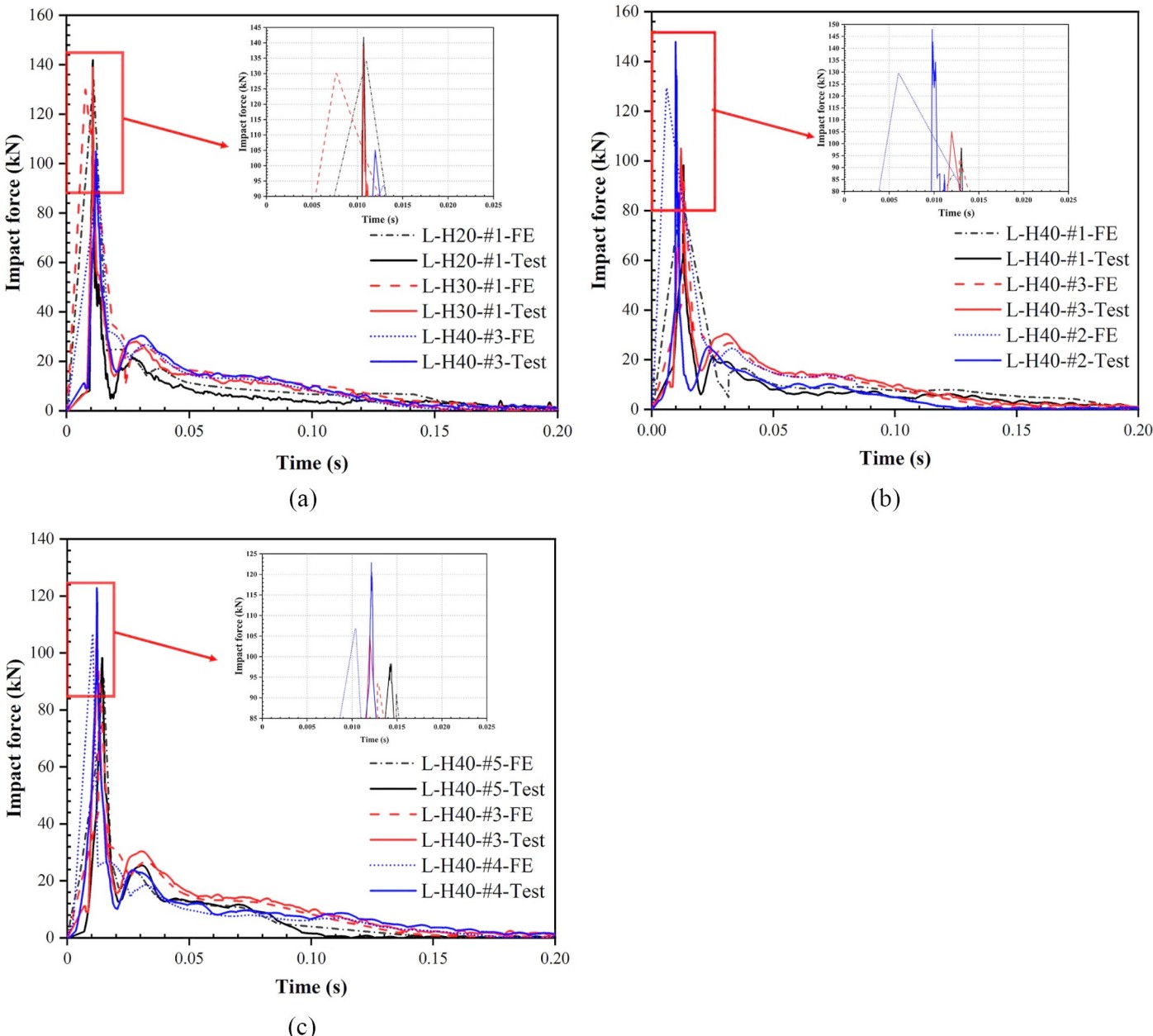

**Fig 10. History curves of impact force for LSFC composite slabs.** (a) Different foam concrete thicknesses; (b) Different foam concrete densities; (c) Different drop heights.

Fm are increased. However, for the thickness of the foam concrete, the patterns of Fmax and Fm change inversely as the thickness of the foam concrete is increased. This is because the resistance of the specimen is positively correlated with the thickness of the foam concrete, and as the thickness is increased, Fm representing the true resistance of the structure will also be increased. However, in the case that the density of the specimens is both 600 kg/m3, with the increase of the thickness of the foam concrete, the contribution of the foam concrete in cushioning is much more significant than its contribution in stiffness. The impact force is effectively reduced through the densification deformation of the foam concrete.

**Table 4. Summary of impact force and displacement test results.**

| Specimen | $F_{max}$ (kN) | $F_m$ (kN) | $D_c$ (kN) | $D_e$ (kN) | $D_q$ (kN) |
|---|---|---|---|---|---|
| L-H40-#1 | 98.27 | 15.31 | 182.22 | 160.25 | 120.47 |
| L-H40-#2 | 147.87 | 26.17 | 147.10 | 130.97 | 68.51 |
| L-H20-#1 | 141.88 | 14.38 | 189.76 | 154.23 | 114.74 |
| L-H30-#1 | 138.90 | 23.21 | 182.53 | 144.40 | 82.16 |
| L-H40-#3 | 105.05 | 24.15 | 172.89 | 140.13 | 69.39 |
| L-H40-#4 | 122.87 | 31.28 | 186.90 | 145.81 | 100.91 |
| L-H40-#5 | 98.25 | 19.68 | 107.09 | 92.23 | 59.30 |

Note: $F_{max}$ is the peak impact force, $F_m$ is the post-peak mean force, $D_c$ is the peak displacement of the center, $D_e$ is the peak displacement of the mid-span edge, $D_q$ is the peak displacement of 1/4 span.

In order to be able to better express the relationship between impact force and support reaction force for LSFC composite slabs under impact loading, the overall trends of impact force and support reaction force were compared and analyzed. The history curves of the support reaction force for LSFC composite slabs are given in Fig 11. By observing and comparing Figs 10 and 11, it can be found that the support reaction force was almost 0 with no significant change during the process of the impact force reaching the peak value from 0. This indicated that the LSFC composite slab at this stage was in an unbalanced state, the response of the force at the support lagged behind the mid-span position, and the impact force mainly produced local indentation damage in the mid-span. With the rapid decrease of the impact force from the peak value, the support reaction force rapidly rose to the peak value, and at the same time, it can be observed that the subsequent fluctuation amplitude and number of fluctuations of the support reaction force were significantly larger than the amplitude and number of fluctuations of the impact force, which can be attributed to the contact area between the bearing and the LSFC composite slab in the rebound or vibration of the case of the change was more intense, resulting in the transmission of the support reaction force was not stable. With the decreased amplitude and number of curve fluctuations, both the impact force and the support reaction force began to decline, and the difference between the two gradually decreased and tended to equilibrate, and eventually the impact force and the support reaction force fell to 0, the end of the deformation of the LSFC composite slab.

### 3.4. Displacement response

The center displacement history curves for different foam concrete thicknesses, densities, and drop heights are given in Fig 12. Table 4 summarizes each specimen's peak displacements of the center, mid-span edge and 1/4 span. The results show that increasing the thickness of foam concrete from 20 mm to 30 mm and 40 mm reduced the peak center displacement by 3.81% and 8.89%, respectively. Increasing the density of foam concrete from 450 kg/m$^3$ to 600 kg/m$^3$ and 750 kg/m$^3$, the center peak displacement was reduced by 5.1% and 19.27%, respectively. This shows that by increasing the thickness and density of the foam concrete, the deformation of the LSFC composite slab can be effectively reduced and the impact resistance of the LSFC composite slab can be improved. When the drop height was increased from 2 m to 2.4 m and 2.8 m, the center peak displacement of LSFC composite slab was increased by 61.44% and 74.53%, respectively, and more significant deformation occurred. The history curves of the center displacement for LSFC composite slabs had a greater slope in the rising stage compared to the conventional SCS composite structures. This is due to the lower overall stiffness of the LSFC composite slab, which allowed it to respond more quickly to external impact force with

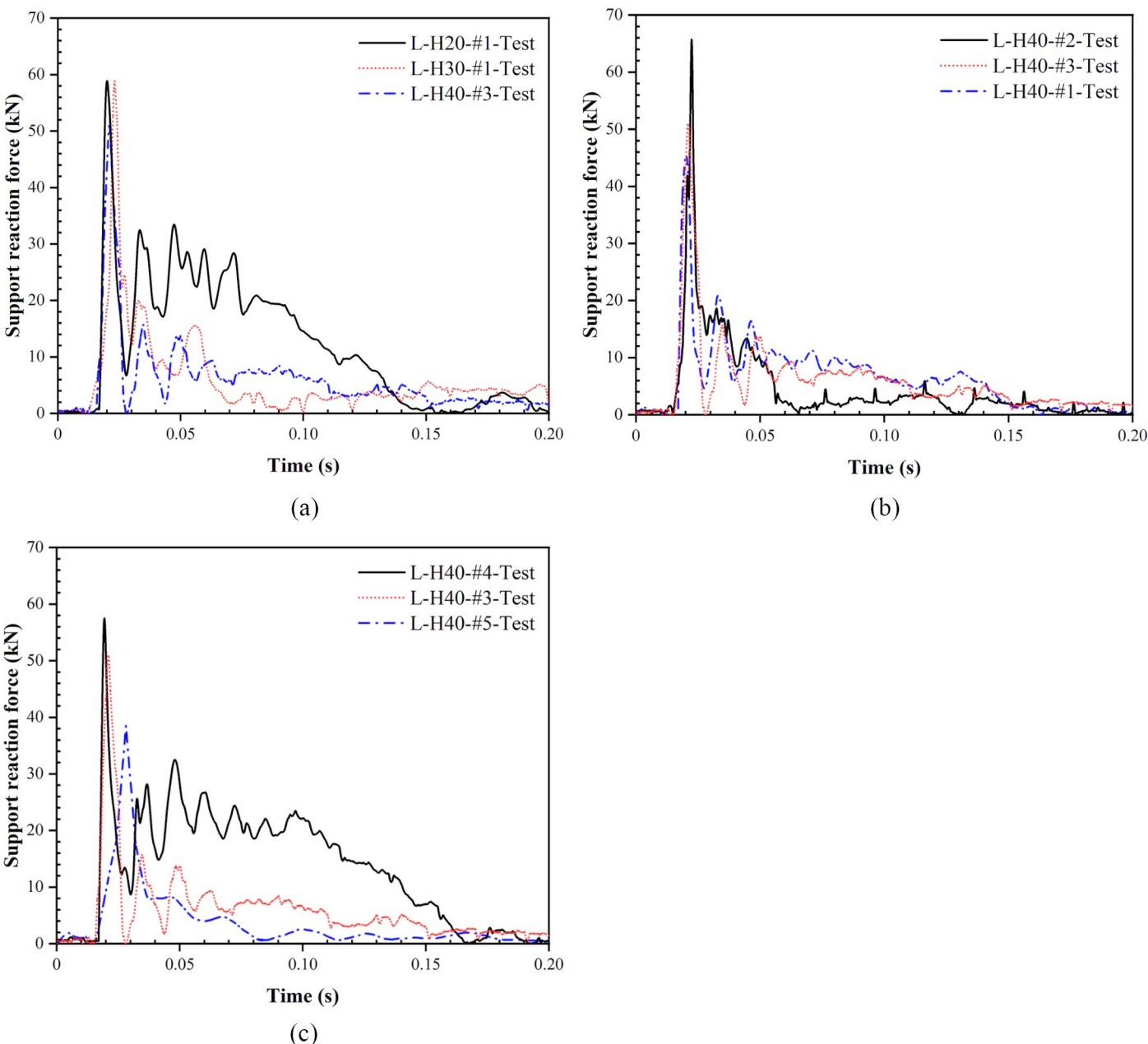

**Fig 11. History curves of support reaction force for LSFC composite slabs.** (a) Different foam concrete thicknesses; (b) Different foam concrete densities; (c) Different drop heights.

greater displacement. In the falling stage, the center displacement history curve had less fluctuation, which can be attributed to the fact that the LSFC composite slab had a better energy absorption effect. When the hammer blow acts on the slab surface, the foam concrete was able to absorb most of the impact energy, which made the rebound kinetic energy of the hammer attenuated. At the same time, due to the lightweight nature of the foam concrete, it was able to reduce the overall mass of the slab, which further reduced the inertia of the fluctuations. The permanently deformed shapes of LSFC composite slabs are shown in Fig 13.

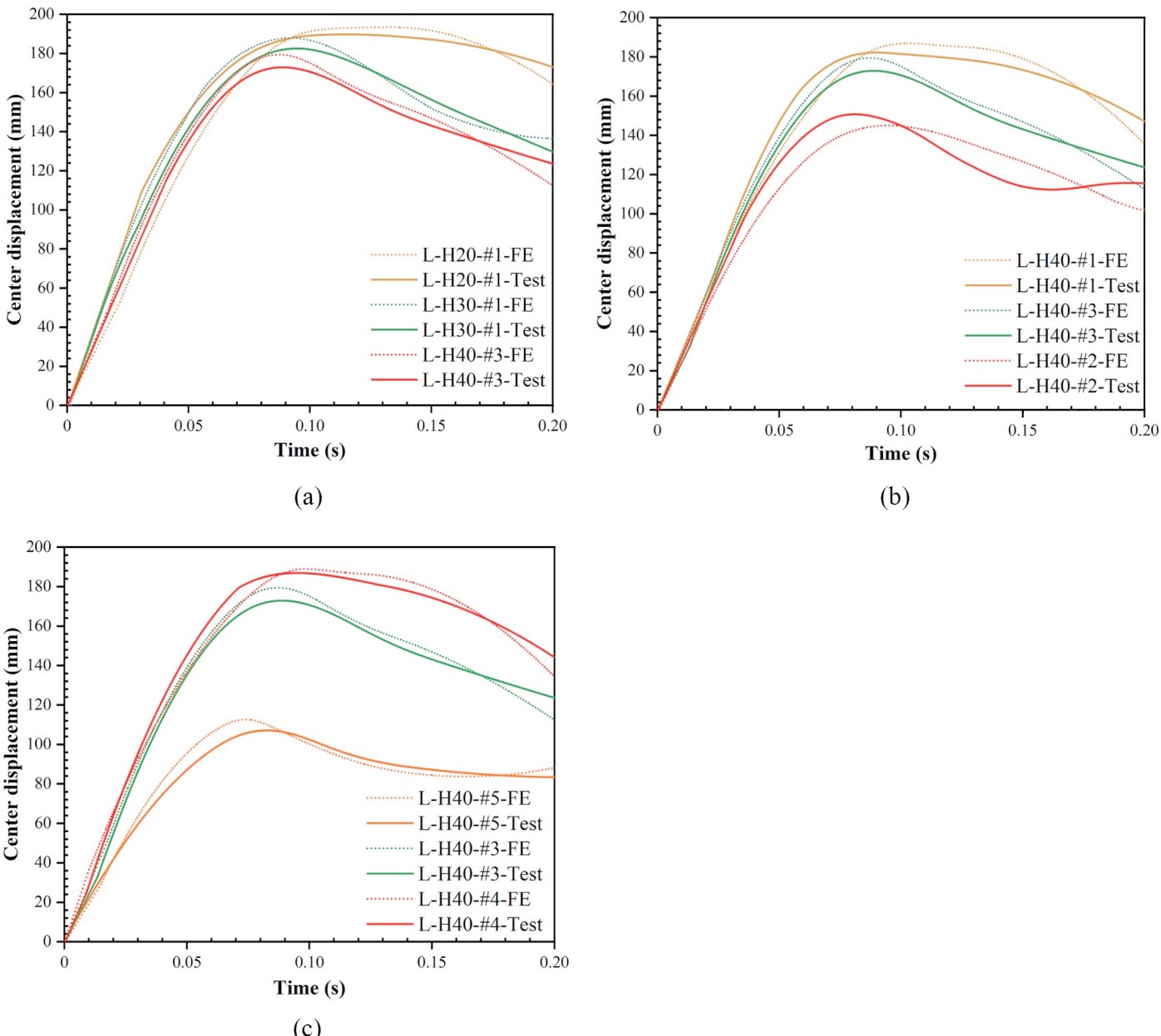

**Fig 12. History curves of center displacement for LSFC composite slabs.** (a) Different foam concrete thicknesses; (b) Different foam concrete densities; (c) Different drop heights.

## 3.5. Energy absorption response

The energy absorption-displacement curves were obtained by integrating the impact force-displacement curves of each specimen, as shown in Fig 14. It can be observed that the overall fluctuation of the energy absorption-displacement curve was slight and roughly shows a linear variation. In addition, because the specimen had slight elastic recovery after the impact load, a small part of the energy absorbed by the specimen will be rereleased in the form of elastic recovery, which made the energy absorption-displacement curve fell back at the end. The specimen's energy absorption rate was increased with the foam concrete thickness for the same

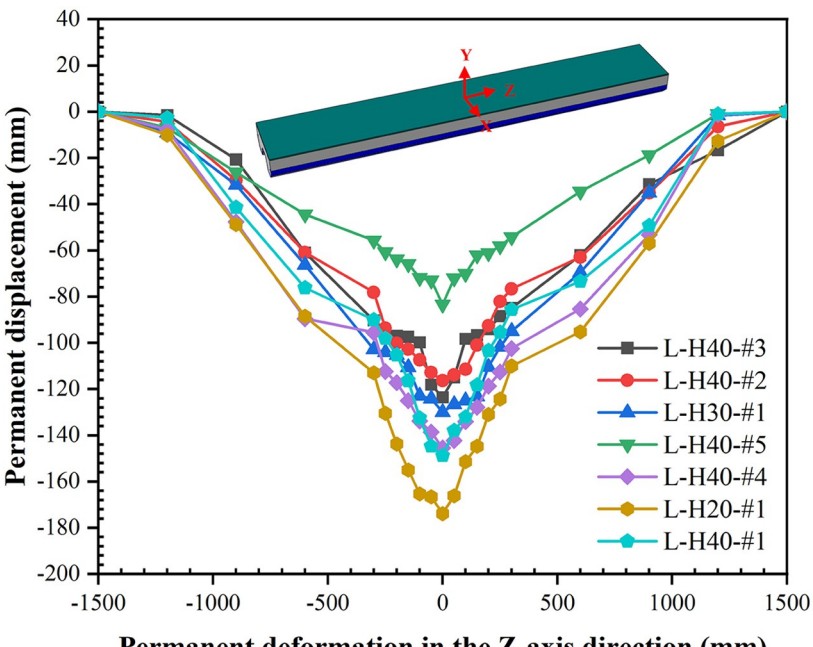

**Fig 13. Permanently deformed shapes of LSFC composite slabs.**

impact energy. However, the energy absorption rate of the specimens did not show a monotonically increasing trend as the density of the foam concrete was increased. This is because the foam concrete layer of L-H40-#1 completely was broke under the impact force, reaching the maximum energy absorption value. With the increase in density, the foam concrete layer of L-H40-#3 increased the stiffness while maintaining a higher porosity, so the energy absorption rate was higher. The energy absorption rate of L-H40-#2 was reduced because the foam concrete at this density became more dense, reducing the pore structure and compressibility, and resulting in more impact energy being bounced. This indicates that foam concrete has a critical density, which has both high deformation resistance and the best energy absorption effect. The energy absorption rate was increased significantly when the drop height was increased from 2 m to 2.4 m and 2.8 m. Still, the difference between the energy absorption rate of L-H40-#3 and L-H40-#4 was tiny, indicating that the energy absorption rate of L-H40-#3 was close to the limit value. Specific energy absorption (*SEA*) is an important index to measure the energy absorption characteristics of LSFC composite slab. Table 5 lists all specimens' energy absorption rates and specific energy absorption values. It can be found that L-H40-#3 had a higher specific energy absorption value and better impact resistance under the same impact energy.

## 4. Numerical simulation study

### 4.1. Setup of finite element model

Due to the great stability of the dynamic explicit algorithm in terms of computational convergence, it is suitable for solving short-time nonlinear problems such as shocks and explosions. Therefore, this paper uses commercial finite element software ABAQUS/Explicit to simulate low-velocity impact tests of LSFC composite slabs. The 3D model is shown in Fig 15. The four-node shell element S4R is used for the W-shaped steel plate, and the eight-node solid element C3D8R is used for the foam concrete, OSB, hammer, support beam and pressure beam.

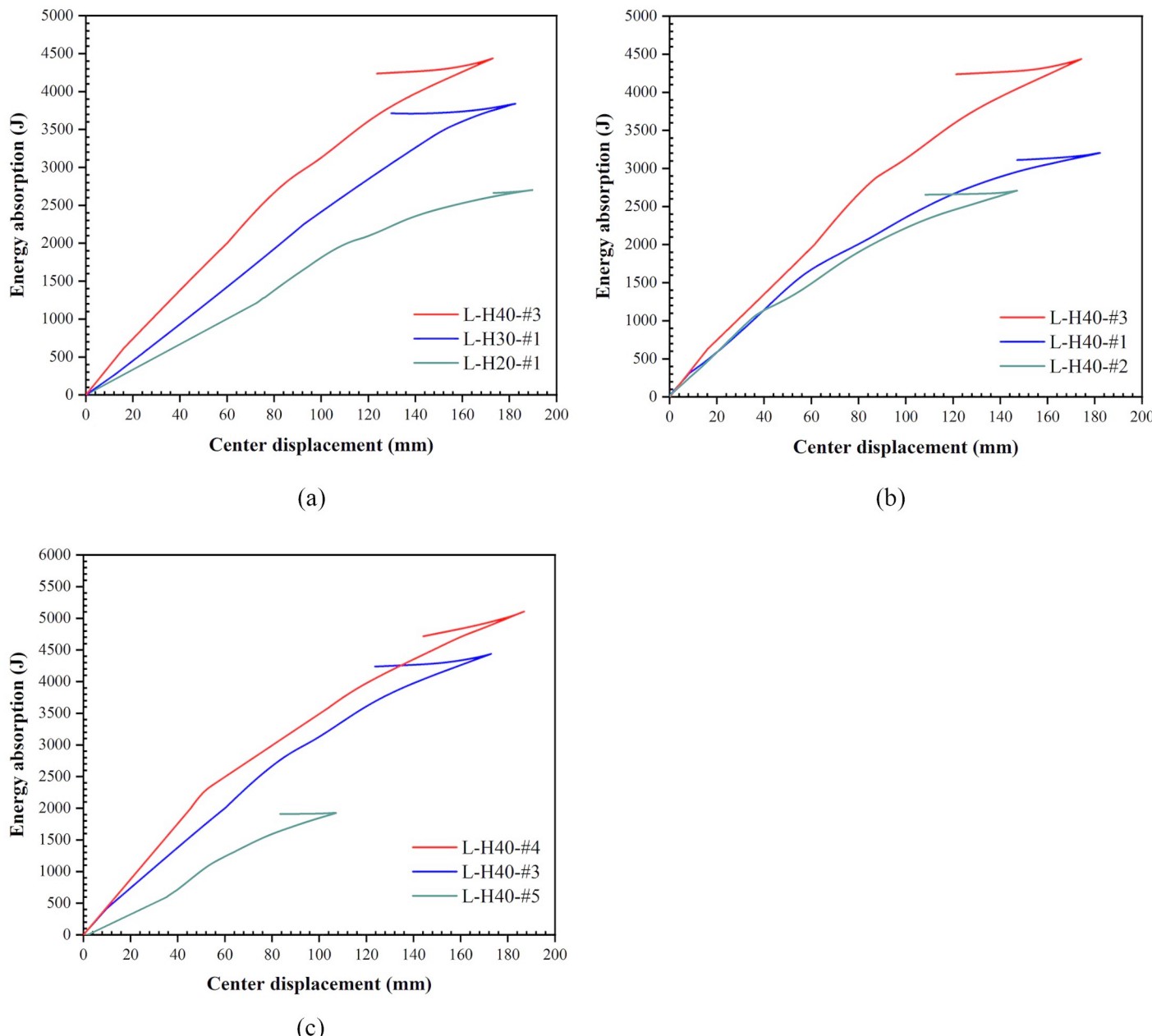

**Fig 14. Energy absorption curves for LSFC composite slab.** (a) Different foam concrete thicknesses; (b) Different foam concrete densities; (c) Different drop heights.

Considering the calculation time and accuracy, the element size of the region of 110 mm each in the left and right of the mid-span is set to 5 mm, and the size of the rest of the elements is set to 10 mm.

In the dynamic analysis, the ideal elastic-plastic constitutive model is used for OSB, the stress-strain relation diagram is shown in Fig 16.

**Table 5. Summary of energy consumption results.**

| Specimen | $E_1$ (J) | $E_2$ (J) | $(E_1/E_2)\times100\%$ | SEA (J/kg) |
|---|---|---|---|---|
| L-H40-#1 | 3111.76 | 5409.60 | 57.52% | 29.61 |
| L-H40-#2 | 2655.44 | 5409.60 | 49.09% | 19.36 |
| L-H20-#1 | 2662.21 | 5409.60 | 49.21% | 24.83 |
| L-H30-#1 | 3713.37 | 5409.60 | 68.64% | 32.23 |
| L-H40-#3 | 4237.64 | 5409.60 | 78.34% | 33.99 |
| L-H40-#4 | 4715.28 | 6311.20 | 74.71% | 38.01 |
| L-H40-#5 | 1908.39 | 4508.00 | 42.33% | 15.27 |

**Note:** $E_1$ is the energy absorption value of LSFC composite slab, $E_2$ is the initial kinetic energy of hammer, SEA is the specific energy absorption value of LSFC composite slab.

The constitutive equation of foam concrete adopts the two-stage formula [36],

$$\begin{cases} y = 2x - 1.1x^2 - 1.7x^3 + 3.8x^4 - 2x^5, \ 0 \le x \le 1 \\ y = (5.2 + 3.7x)/(1 + 7.9x), x > 1 \end{cases} \quad (2)$$

$x = \varepsilon/\varepsilon_{pr}$, $y = \sigma/f_{pr}$, $\varepsilon_{pr}$ is the peak strain, $f_{pr}$ is the peak intensity.

The five-stage elastoplastic constitutive model was used to simulate the low-carbon steel, the stress-strain curves are shown in Fig 17 [37–39]. Considering the strain rate effect, the C-S (Cowper-Symonds) model was used to calculate the yield stress at different strain rates as shown in Eq (3). The values of $D$ and $P$ were taken as 6844 s$^{-1}$ and 3.91, respectively [40,41]. Since the W-shaped steel plates did not fracture after the test, the fracture behavior of the W-shaped steel plates in the FE model was not considered.

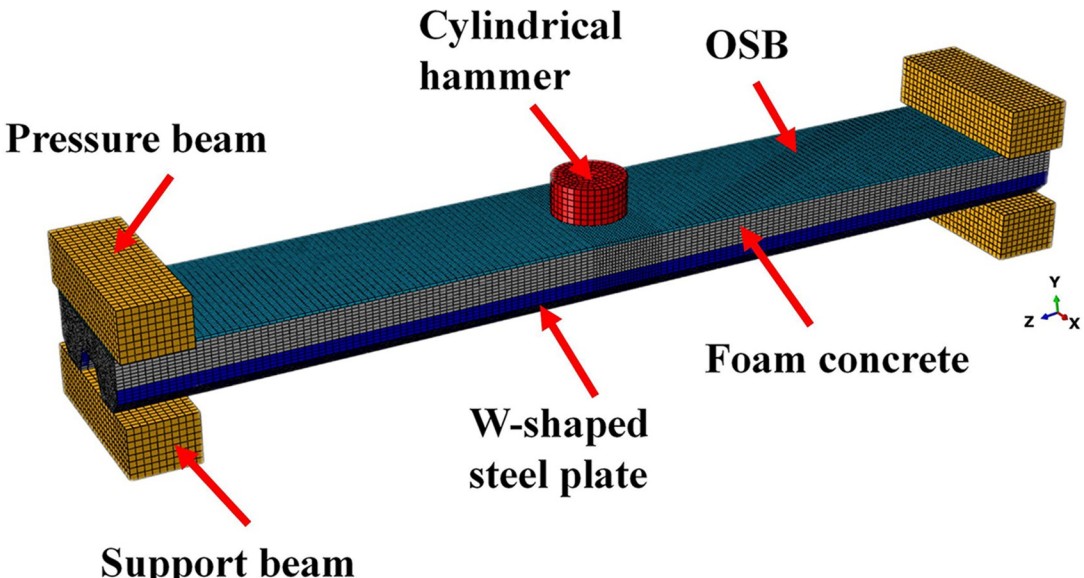

**Fig 15. LSFC composite slab finite element model.**

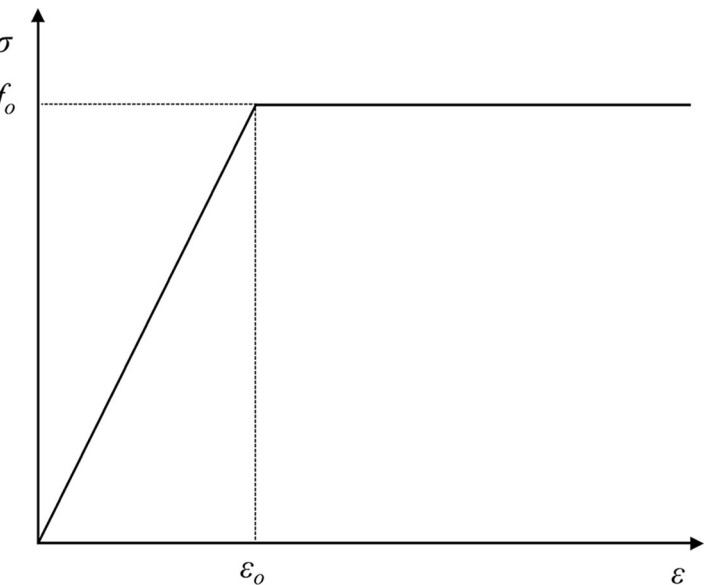

**Fig 16. Stress-strain relation diagram of OSB.**

$$f_y^d/f_y = 1 + (\dot{\varepsilon}/D)^{1/P} \tag{3}$$

$f_y^d$ is the dynamic yield strength under $\dot{\varepsilon}$; $f_y$ is the static yield strength; $\dot{\varepsilon}$ is the strain rate.

Considering that there was no significant separation between the W-shaped steel plate and the foam concrete during the impact process, the contact command between the W-shaped steel plate and the foam concrete was set to tie in order to simplify the model. The flange plate of the W-shaped steel plate is the embedded region, and the foam concrete is the host region.

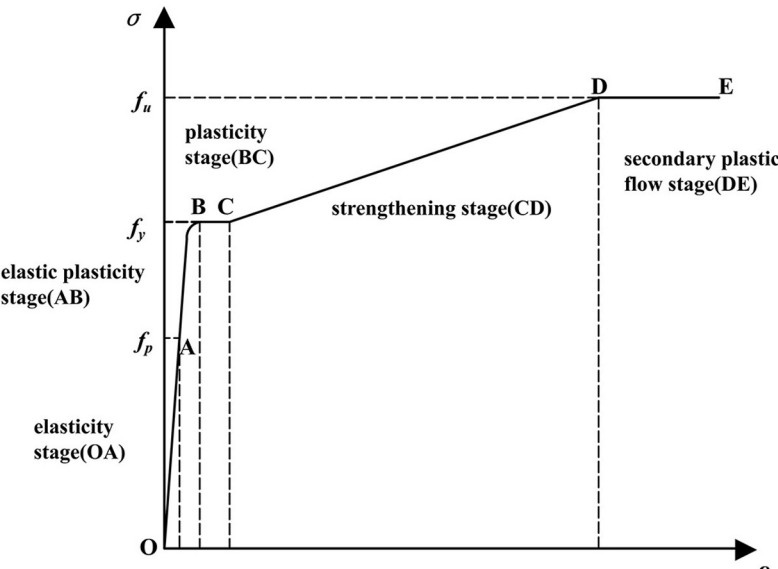

**Fig 17. Stress-strain relation diagram of steel.**

Embedding the flange plate of the W-shaped steel plate in the foam concrete. The hammer is bound to the reference point, and a mass of 230 kg is assigned to the reference point. Restrict all degrees of freedom of the reference point except the vertical direction (Y direction). Give the hammer an initial velocity in the predefined field-corresponding to the drop height ($v = \sqrt{2gh}$, $h$ is the drop height). The hammer is located at the center of the OSB, and surface-to-surface contact is used between the LSFC composite slab and the hammer. In the contact properties, the normal behavior was set to hard contact and the tangential behavior was set to frictionless. Simulation of the cementation between layers of OSB and foam concrete by defining the cohesive behavior. The parameters related to cohesive behavior are shown in Table 6. Tie commands were used between the support beams and the W-shaped steel plates, and between the pressure beams and the OSB. Since the boundary conditions in the test were simply supported boundaries, the nodes of the support beam and pressure beam in the FE model were constrained only in the vertical and horizontal directions without restricting the rotation of the nodes. In addition, to improve the computational efficiency, the hammer, the support beams and the pressure beams were set as rigid body models without considering their deformations during the impact process.

## 4.2. Finite element results and analysis

Fig 18 gives the vertical permanent deformation of specimen L-H40-#2 obtained from the prediction of the FE model, and the damage characteristics of the test and the FE model were also compared. The results show that the setup FE model can predict the overall and local deformation characteristics of the LSFC composite slabs. The combined deformation pattern of the slabs obtained by the FE model was consistent with the test results. In addition, the FE model can also capture the interlayer separation between OSB and foam concrete. In order to visualize the overall deformation of the slabs, the vertical permanent deformation of the W-shaped steel plate is given in Fig 18B. The peak impact force and peak displacement obtained from the FE model and the tests were compared and are shown in Table 7. It was found that the deviation of the prediction of the peak impact force by the FE model was not more than 13%, and the deviation of the prediction of the peak displacement at the center was not more than 5%. The above analysis proves that the FE model has high accuracy.

Fig 19 shows the stress and strain diagrams for W-shaped steel plates at the bottom of all specimens. Under the action of impact load, the LSFC composite slab is bent and deformed, and the W-shaped steel plate is mainly subjected to tension along the Y-axis direction. Due to the restraining effect at the support beams, the stress concentration occurs in the local area of the W-shaped steel plate. At the same time, it can be found that the equivalent plastic strain was increased significantly near the impact point of LSFC composite slabs and at the junction of W-shaped steel plates and supports.The equivalent plastic strain of W-shaped steel plates was mainly distributed in the mid-span region and was decreased gradually from the mid-span along the spanwise direction, but it was increased again at the supports. The increase of equivalent plastic strain at the support was due to the overall bending of the LSFC composite slab during the impact process, which leaded to the crease of the end of W-shaped steel plates at the junction with the support.

## 5. ESDOF system deflection calculation method

### 5.1. Setup of the ESDOF model

As shown in Fig 20, the ESDOF model consists of the hammer mass ($m_h$), the effective mass of the LSFC composite slab ($m_s$), the effective mass of the model($m_e$), the impact velocity ($v_0$), the

**Table 6. Properties for the cohesive behavior [42].**

| $\sigma$ (MPa) | $\tau_1$ (MPa) | $\tau_2$ (MPa) | $K_{nn}$ (N/mm³) | $K_{tt,1}$ (N/mm³) | $K_{tt,2}$ (N/mm³) | $\delta_f$ (mm) |
|---|---|---|---|---|---|---|
| 5 | 1.5 | 1.5 | 500 | 500 | 500 | 1.5 |

**Note:** $\sigma$ is strength in the normal direction, $\tau_1$ and $\tau_2$ are shear strengths, $K_{nn}$ is stiffness in the normal direction, $K_{tt,1}$ and $K_{tt,2}$ are stiffness in the shear directions, $\delta_f$ is displacement at failure.

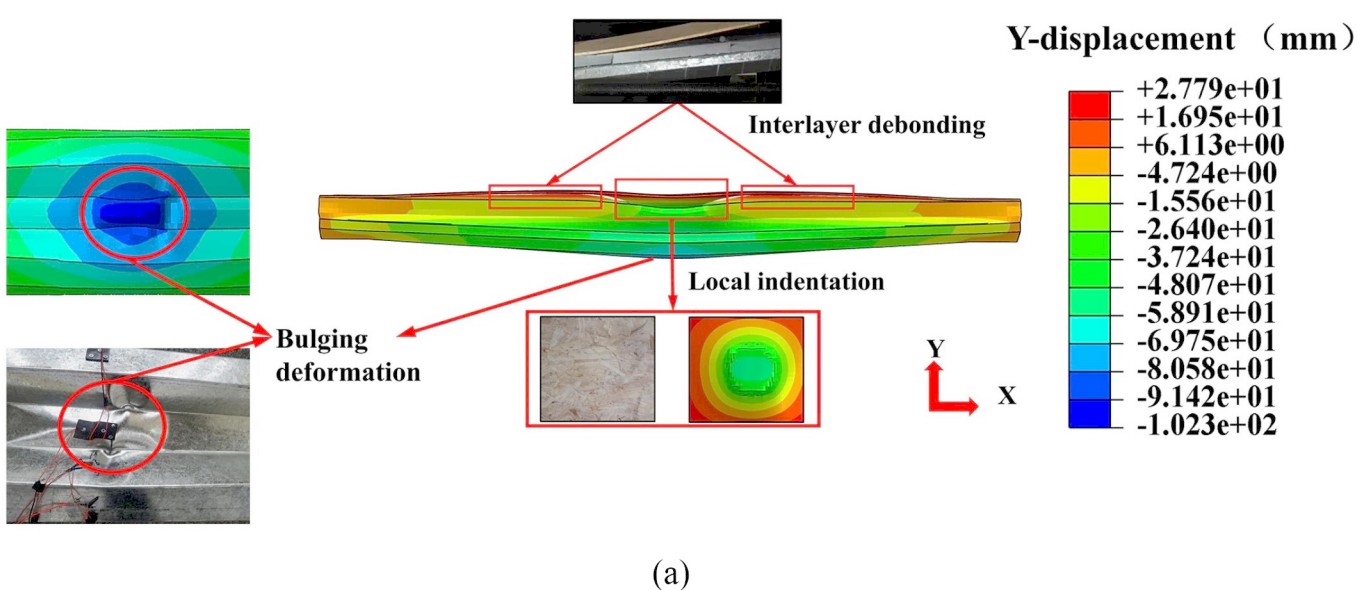

(a)

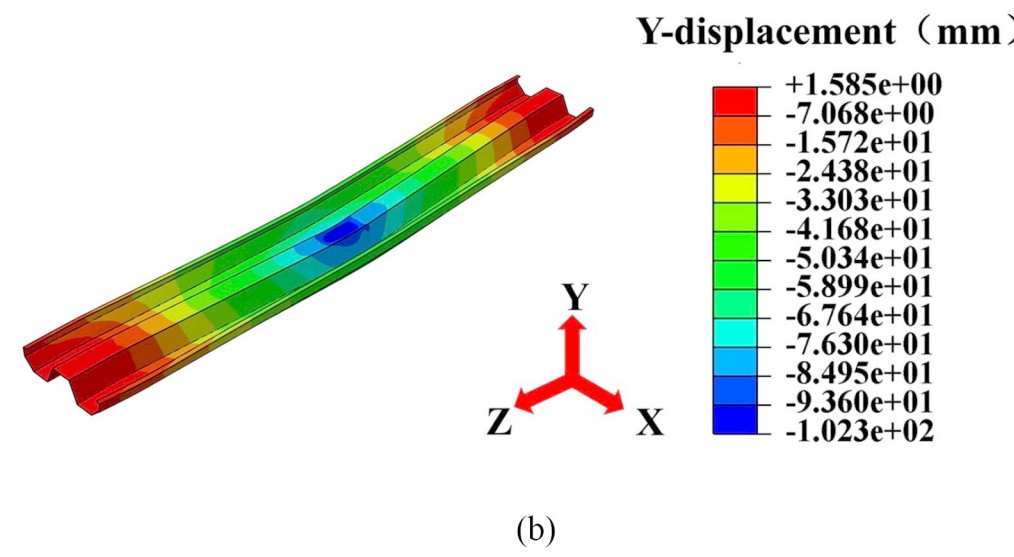

(b)

**Fig 18. Vertical permanent displacement diagram of L-H40-#2.** (a) Vertical permanent displacement of LSFC composite slab; (b) Vertical permanent displacement of W-shaped steel plate.

**Table 7. Comparison of FE simulation and test results.**

| Specimen | $F_{max}$ (kN) | $F_{maxFE}$ (kN) | $F_{maxFE}/F_{max}$ | $D_c$ (mm) | $D_{cFE}$ (mm) | $D_{cFE}/D_c$ |
|---|---|---|---|---|---|---|
| L-H40-#1 | 98.27 | 87.06 | 0.89 | 182.22 | 186.05 | 1.02 |
| L-H40-#2 | 147.87 | 129.98 | 0.88 | 147.10 | 146.54 | 1.00 |
| L-H20-#1 | 141.88 | 134.85 | 0.95 | 189.76 | 193.08 | 1.02 |
| L-H30-#1 | 138.90 | 130.70 | 0.94 | 182.53 | 187.62 | 1.03 |
| L-H40-#3 | 105.05 | 94.36 | 0.90 | 172.89 | 179.23 | 1.04 |
| L-H40-#4 | 122.87 | 107.50 | 0.87 | 186.90 | 188.72 | 1.01 |
| L-H40-#5 | 98.25 | 91.83 | 0.93 | 107.09 | 111.98 | 1.05 |

**Note:** $F_{max}$ is the peak impact force obtained from the test, $F_{maxFE}$ is the peak impact force obtained from the FE simulation, $D_c$ is the peak displacement of the center point obtained from the test, $D_{cFE}$ is the peak displacement of the center point obtained from the FE simulation.

displacement of the model ($u_t$) and the structural stiffness of the model ($k$). The motion equation of the ESDOF model is as follows:

$$m_e \ddot{u}(t) + ku(t) = 0 \tag{4}$$

## 5.2. Effective mass of the structure

In the ESDOF model, $m_h$ and $m_s$ can be derived from the experimental design. The $m_e$ is not a simple sum of the two masses but requires multiplying $m_s$ by $k_m$. The $k_m$ in this paper uses 0.2 for the elastic stage and 0.1 for the plastic stage [43], as shown in Eq (5),

$$m_e = k_m m_s + m_h \tag{5}$$

where $k_m$ is the quality conversion factor.

## 5.3. Stiffness

**5.3.1. Tangential stiffness at the yield point.** The overall central deflection of the bottom surface is equal to the sum of the global bending deflection and the local bulging deflection. Based on Hooke's law, the tangential stiffness of the LSFC composite slab is given by Eqs (6) and (7):

$$F_s = k_L \delta_L = k_p \delta_p = k_s \left( \delta_L + \delta_p \right) \tag{6}$$

$$k_s = \frac{k_L k_p}{k_L + k_p} \tag{7}$$

where $F_s$ is the spring element resistance, $\delta_L$ is the global bending deflection, $\delta_P$ is the local bulging deflection, $k_s$ is the tangential stiffness at the yield point, $k_L$ is the global bending stiffness, and $k_p$ is the local bulging stiffness.

(1) Global bending stiffness

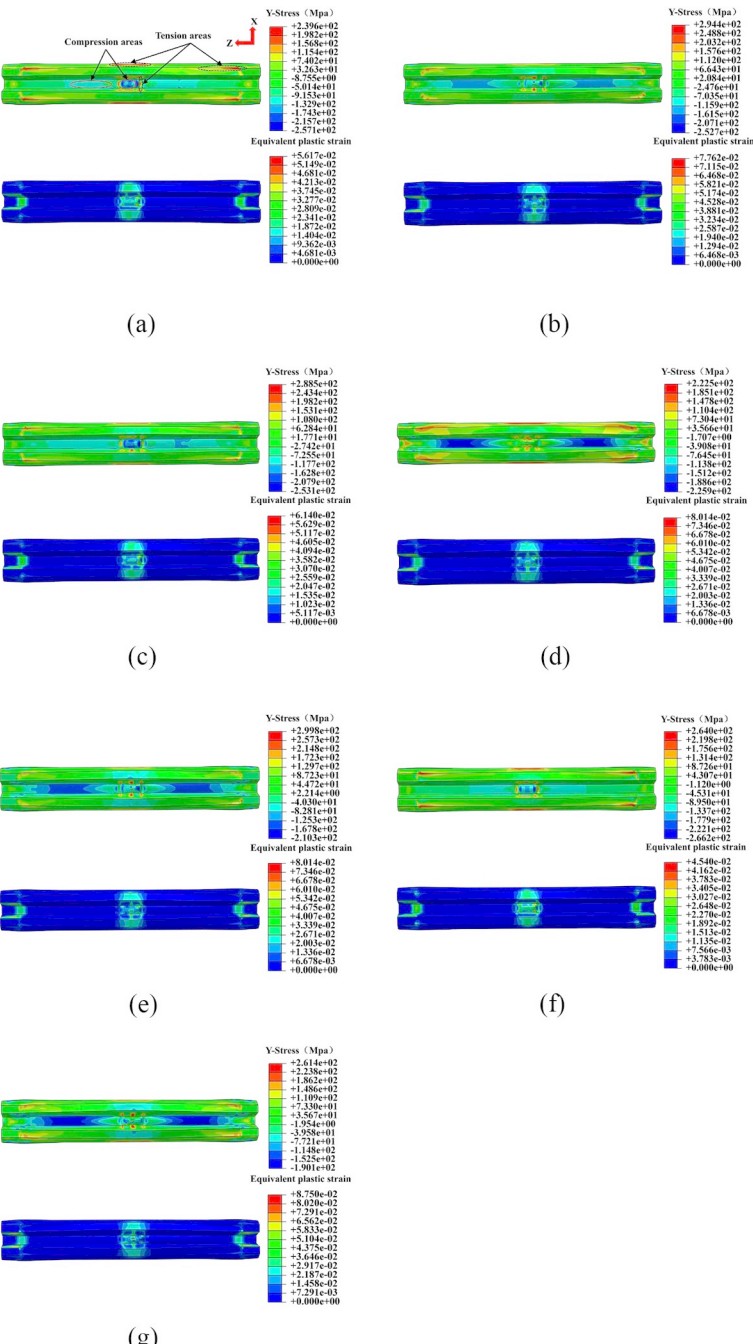

**Fig 19. Stress and strain diagram of W-shaped steel plate.**

Referring to the plate and shell theory [44], the global bending stiffness of LSFC composite slabs in the range of elastic deformation is derived, as shown in Eq (8):

$$k_L = \frac{D_{sf} + D_o}{0.01695L_s^2} \tag{8}$$

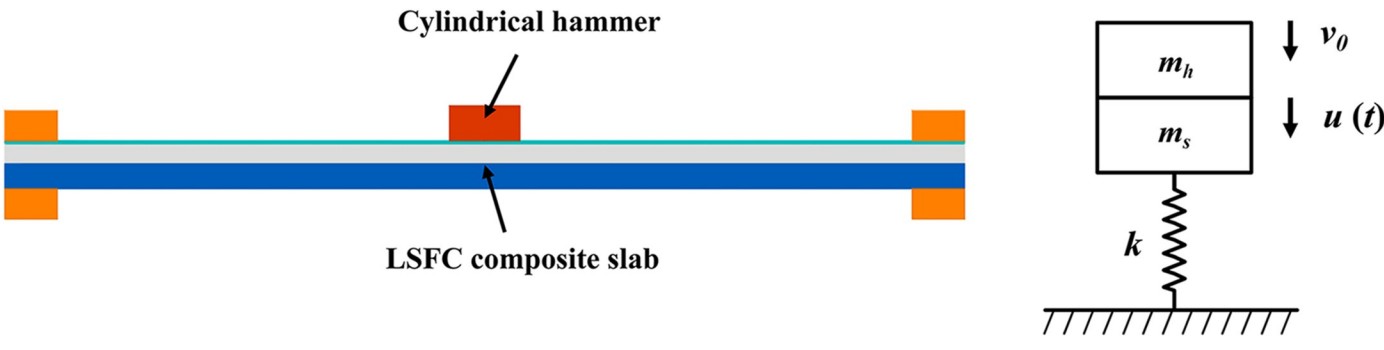

**Fig 20. ESDOF model.**

$D_{sf}$ and $D_o$ are the bending stiffnesses of steel plate-foam concrete composite slab and OSB, respectively. Among them, the bending stiffness of the steel plate-foam concrete composite slab is referred to in the study by Liu [45]. The equation is as follows:

$$D_{sf} = \frac{E_s A_s h_0{}^2}{(1.15\psi + 0.2 + 6\alpha_E \rho)L_s} \tag{9}$$

$$D_o = \frac{ab^3 E_0}{12L_s} \tag{10}$$

where $E_s$ is the elastic modulus of W-shaped steel plate, $A_s$ is the cross-sectional area of steel plate, $h_0$ is the distance from the neutral axis of W-shaped steel plate to the top surface of the foam concrete, $\psi$ is the uniform coefficient of tensile strain at the center of gravity of the profiled steel plate between cracks, taken as 0.2, $\alpha_E = E_s/E_c$, $\rho$ is the calculation of the steel content of the composite slab in the cross-section, $L_s$ is the length of the clear span, $a$ is the width of OSB section, and $b$ is the height of OSB section.

Since the yield strength of foam concrete is greatly lower than that of W-shaped steel plate, foam concrete has already crumbled when W-shaped steel plate yields. Therefore, the reduction factor for the yield point foam concrete stiffness is 0.7 [46], and the equation is as follows:

$$D_L = 0.7D_{sf} + D_o \tag{11}$$

(2) Local bulging stiffness

At present, there is no relevant formula to describe the local bulging stiffness of W-shaped steel plate. Therefore, in this paper, based on the study of Sohel and Guo [7,16], the local bulging stiffness of W-shaped steel plate is derived as shown in Eq (12):

$$k_p = 4\pi f_y t_s \tag{12}$$

**5.3.2. Tangential stiffness after yielding.**    After yielding, the overall stiffness of the plastic stage is reduced to 0.15 times that of the elastic stage [47], and the tangential stiffness of this stage is expressed in Eq (13):

$$k_{sp} = 0.15k_s \tag{13}$$

## 5.4. Numerical solution of the equations of motion

$$m_e = \begin{cases} 0.2m_s + m_h, & 0 \le u(t) \le u_y \\ 0.1m_s + m_h, & u(t) > u_y \end{cases} \tag{14}$$

where $u_y$ is yield displacement of W-shaped steel plate.

The initial conditions of the equations of motion are:

$$u(0) = 0 \tag{15}$$

$$\dot{u}(0) = \frac{m_h v_0}{m_e} \tag{16}$$

The finite difference method is used to solve the equations of motion. The equations for the displacement, velocity and acceleration at the time $t_i$ are:

$$u(t_i) = u_i \tag{17}$$

$$\dot{u}(t_i) = \dot{u}_i = \frac{u_{i+1} - u_i}{\Delta t}$$

$$\ddot{u}(t_i) = \ddot{u}_i = \frac{u_{i+1} - 2u_i + u_{i-1}}{\Delta t^2}$$

where $\Delta t = 5 \times 10^{-5}$ s is the time step. Eq (4) can be written as:

$$m_e \frac{u_{i+1} - 2u_i + u_{i-1}}{\Delta t^2} + ku_i = 0 \tag{18}$$

The displacement at the time $t_{i+1}$ is further solved by the recurrence relation, as shown in Eq (19):

$$u_{i+1} = 2u_i - u_{i-1} - \frac{\Delta t^2 k u_i}{m_e} \tag{19}$$

The first two terms of the recurrence relation are:

$$u_0 = 0 \tag{20}$$

$$u_1 = \frac{u_0 + \dot{u}_0 \Delta t}{a} = \frac{m_h v_0 \Delta t}{a m_e} \tag{21}$$

where $a$ is the amplification factor of the second term, and $a$ is taken as 0.0018.

The history curves of center displacement obtained from the tests and ESDOF model are shown in Fig 21. The results show that the predicted results of the ESDOF model fit well with the test results. The maximum prediction error of peak center displacement and residual displacement is 9.0%, which is within the acceptable range. The applicability of the proposed ESDOF model in predicting the displacement response of LSFC composite slabs is proved.

## 6. Conclusions

In this paper, the dynamic response of LSFC composite slabs under impact load is studied by means of tests, numerical simulation and theoretical analysis. The main conclusions are summarized as follows:

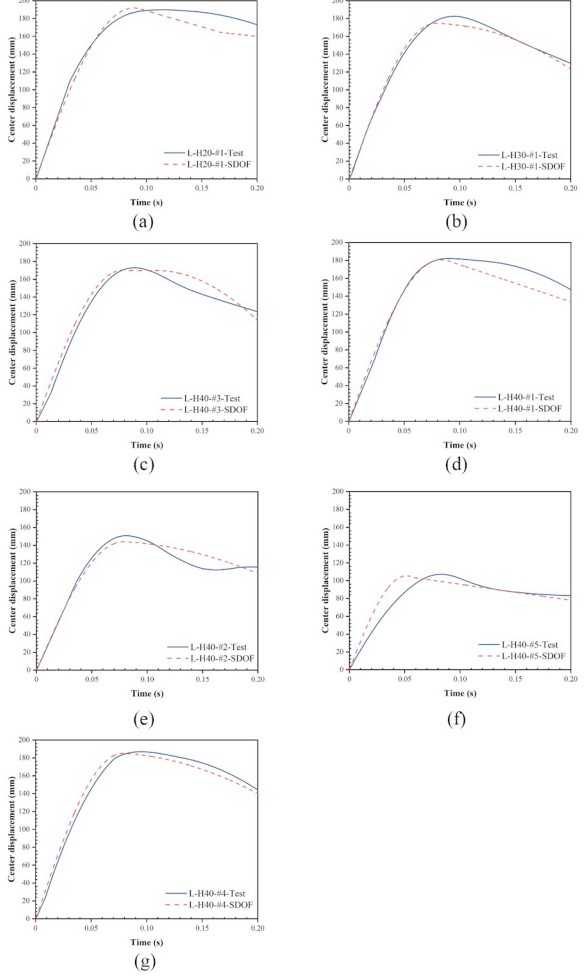

**Fig 21. History curves of center displacement for ESDOF model and test.**

1. Under impact loading, the damage mode of LSFC composite slabs consists of overall bending and local indentation. The impact process is divided into three phases: inertial, loading, and unloading stages.

2. The maximum deformation of the LSFC composite slab can be reduced by increasing the density and thickness of the foam concrete. When the thickness of foam concrete was increased from 20 mm to 40 mm, the maximum displacement of the LSFC composite slab was decreased by 8.89%. When the density of foam concrete was increased from 450 kg/m³ to 750 kg/m³, the maximum displacement of the LSFC composite slab was decreased by 19.27%.

3. Increasing the thickness of foam concrete can increase the energy absorption rate of LSFC composite slabs. However, the energy absorption rate of LSFC composite slabs peaked when the foam concrete density was 600 kg/m³. This indicates that there is an energy absorption critical value for LSFC composite slabs as the density of foam concrete increases. When the density of foam concrete reaches this critical value, the energy absorption rate starts to decrease.

4. The maximum difference between the impact force and displacement history curves obtained by the FE model and the test results is not more than 13%, which is in good agreement. Meanwhile, the distribution characteristics of stress and strain of W-shaped steel plates are revealed.

5. The ESDOF model was set up, and the maximum difference in the predicted displacement response of the LSFC composite slab is not more than 9%, which proves that the prediction results of the ESDOF model are reasonable.

## Supporting information

**S1 Fig. Mesh stability verification of impact force.**
(TIF)

**S2 Fig. Mesh stability verification of displacement.**
(TIF)

**S1 File. Minimal data set of impact force and center displacement.**
(XLSX)

## Author Contributions

**Conceptualization:** Linghui Meng, Lei Wang, Qiang Xu.

**Data curation:** Lei Wang, Qiang Xu, Zhenhui Zhang.

**Formal analysis:** Jinbo Chen.

**Methodology:** Linghui Meng, Lei Wang, Jinbo Chen, Qiang Xu.

**Software:** Linghui Meng, Lei Wang, Shuwang Yang.

**Validation:** Lei Wang, Jinbo Chen, Minghao Yang.

**Writing – review & editing:** Linghui Meng, Qiang Xu, Bowen Liu.

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
