## [Decision Letter · Decision Letter 0]

23 Oct 2023

PONE-D-23-29569Impact response of lightweight steel form concrete composite slabs： Experimental, numerical and analytical studiesPLOS ONE

Dear Dr. Xu,

Thank you for submitting your manuscript to PLOS ONE. After careful consideration, we feel that it has merit but does not fully meet PLOS ONE’s publication criteria as it currently stands. Therefore, we invite you to submit a revised version of the manuscript that addresses the points raised during the review process.

We look forward to receiving your revised manuscript.

Kind regards,

Shaker Qaidi

Academic Editor

PLOS ONE

Journal Requirements:

Additional Editor Comments:

The evaluations from the peer reviewers regarding your submitted work have been duly received. Upon reviewing their feedback, it is evident that they recommend that you revise your manuscript. Therefore, the authors should consider each comment and decide on the best course of action for their research.

Reviewers' comments:

Reviewer's Responses to Questions

**Comments to the Author**

1. Is the manuscript technically sound, and do the data support the conclusions?

Reviewer #1: Yes

Reviewer #2: Yes

Reviewer #3: Yes

2. Has the statistical analysis been performed appropriately and rigorously? 

Reviewer #1: N/A

Reviewer #2: Yes

Reviewer #3: N/A

3. Have the authors made all data underlying the findings in their manuscript fully available?

Reviewer #1: No

Reviewer #2: Yes

Reviewer #3: Yes

4. Is the manuscript presented in an intelligible fashion and written in standard English?

Reviewer #1: Yes

Reviewer #2: No

Reviewer #3: Yes

5. Review Comments to the Author

Reviewer #1: Dear authors,

I believe the following suggestions could be considered to enhance the paper.

Major issues:

- Improving significantly the resolution of all the figures, graphics and pictures and enlarging them all.

- Referring to the FE model robustness and stability. Have you checked the stability of the results against mesh refinements? time integration step? element interpolation functions?

- Describing the FE model in much more detail to ensure its reproducibility, primarily if you will not provide the ABAQUS code or files as supplementary material. For example, what does the sentence mean in line 235 “The bond between the W-shaped steel plate and the foam concrete is assumed to be good” in terms of the FE boundary conditions and connection between meshes? Continuous mesh, bonded contact?

- Rewriting all the conclusions by adding the numerical values that support them.

Minor issues:

- Can you please enlarge the font size? It is really hard to read font size 8 when printed.

- Line 51-52. “It can not only […] comprehensive benefit”. It is difficult to understand. Please clarify what you mean.

- Line 54. The word “reasonableness” does not seem common in this context. Do you mean “reliability”?

- Line 59. Please explain what is the “grip force”. Where is that force acting on?

- Section 2.1 Lines 58-65. All parameters included in Table 1 do not need to be described in such detail in the text.

- Section 2.1 Table 1. If possible, please group the rows with the same value for clarity to avoid repeating numbers, especially when only one value, such as the “Impact position” or the hammer mass.

- Section 2.1 Table 1. Please explain the meaning of the specimen codes in the text.

If the first number beside the letter H, is the thickness of the foam concrete, why not use the whole number in mm, L-H40-#1?

What is “L” standing for?

Adding # to the last number would clearly indicate that it is just numbering the specimens, but you should explain that you are numbering the specimens with shared foam thickness, is this right?

- Lines 105, 109. The passive impersonal voice is commonly used in technical papers, not “we plotted” or “we can assume”.

- Line 139. Reference [30] does not refer to the D’Alembert principle but to an article describing the inertia effects. Therefore, please move [30] to the end of the sentence.

- Line 156. Please rewrite the sentence including “…resistance will be stronger, …”. The whole sentence looks a bit redundant.

- Formulae (2) and (3). Maybe it is not necessary to show the formula for the mean or specific values.

- Line 195. Remove the words “to the outside world”, maybe it is not necessary.

- Sections 4.1. and 5.1. “Establishment” is not appropriate to this context. Better use “Setup”

- Section 5. Please, write every single word for ESDOF the first time you write it; that is in line 265, not in line 272, where you can write ESDOF.

Sincerely.

Reviewer #2: This study examines the low-velocity impact response of lightweight steel form concrete composite (LSFC) slabs. Tests reveal that the LSFC composite slabs experience local indentation and overall deformation. The peak impact force is positively correlated with foam concrete density and negatively correlated with foam concrete thickness on the top plate. Optimal energy absorption occurs at a 40mm thickness and 600 kg/m³ density. The study includes a validated finite element model and an equivalent-single-degree-of-freedom (ESDOF) model for initial impact resistance assessment.

Despite the experimental work conducted by the authors, this article has some issues that require a rewrite：

1. The exploration and analysis of the experimental data are not sufficiently profound, which does not align with the effort invested in the experiments.

2. The numerical model used for the analysis is relatively simplistic and does not reference state-of-the-art methods.

3. The introduction of the experimental preparation is not comprehensive enough, and the corresponding figures lack detail.

4. The captions are not clear enough, especially in the inset figures, where the internal scales are entirely illegible, as in Figure 10.

Reviewer #3: The manuscript presents an interesting experimental and numerical study on the impact response of lightweight steel foam concrete composite slabs. The work is technically sound with a good combination of physical testing, FE modeling, and analytical modeling. The results demonstrate the effects of foam concrete thickness and density on the slabs' impact resistance and energy absorption capability. The failure modes, force-displacement responses, and energy absorption characteristics are well characterized. The FE model shows good agreement with experiments. The ESDOF model reasonably predicts displacements. Overall the work provides useful insights into the impact behavior of this novel composite system. The manuscript is well-written and easy to follow. I have provided 18 detailed comments and suggestions aimed at strengthening the work further, clarifying some aspects, and improving the presentation quality. Addressing these should enhance the manuscript's contribution and prospects for publication.

Comments:

1. In the abstract, summarize the key findings and conclusions in a concise manner.

2. In the introduction, provide more background on the limitations of traditional steel-concrete composite slabs in high-rise buildings to motivate the need for the lightweight alternative studied here.

3. In Section 2.1, provide a schematic cross-section of the slab with dimensions labeled to help visualize the geometry.

4. specify how the support conditions simulate a simply supported slab.

5. In Section 3.1, comment on whether the observed failure modes match expectations and any prior work.

6. explain the physical meaning/significance of the dip angle change rate parameter.

7. explain the cause of the small fluctuations seen in the loading portion of the force histories.

8. In Section 3.4, discuss how the displacement results compare with any prior studies on similar composite systems.

9. explain why elastic springback causes the final drop in the energy curves.

10. In Section 4.1, provide more details on the cohesive element properties used between material layers.

11. In Section 4.2, present quantitative validation metrics for the FE model rather than just a qualitative statement of good agreement.

12. define all variables when the ESDOF model is first introduced.

13. In Section 5.3.1, explain the physical meaning of the reduction factor used for foam concrete stiffness.

14. In Section 5.4, explain the selection of the amplification factor.

15. In the conclusions, highlight the major findings more clearly.

16. Carefully proofread the manuscript to fix minor grammar issues.

17. Add a table summarizing all specimen details.

18. Provide more discussion of how the findings can be utilized for design purposes.

6. PLOS authors have the option to publish the peer review history of their article (what does this mean?). If published, this will include your full peer review and any attached files.

Reviewer #1: No

Reviewer #2: No

Reviewer #3: **Yes: **Mahmoud Akees

---

## [Author Response · Author response to Decision Letter 0]

4 Dec 2023

Reviewer #1:

1.Question:

Improving significantly the resolution of all the figures, graphics and pictures and enlarging them all.

Response:

We sincerely appreciate the valuable suggestion. We have improved the resolution of all the figures, graphics and pictures and enlarged them significantly in the manuscript

2.Question:

Referring to the FE model robustness and stability. Have you checked the stability of the results against mesh refinements? time integration step? element interpolation functions?

Response:

Thank you for your rigorous comment. As you have considered, mesh refinements, time integration step, and element interpolation functions are very important for the computation of finite element models. The detailed analyses are as follows:

(1) When meshing, we referred to Wang's[13] study and determined the element size to be 10 mm. At this time, we found that the impact force and midspan displacement curves obtained by the current size were in good agreement with the experimental results. Then, we further set the element size to 5 mm, and the impact force and midspan displacement curves obtained at this time did not change much, but the calculation time was greatly increased. Therefore, to present the local damage better, we used the 5 mm element size in the area of 110 mm left and right of the midspan and the 10 mm element size in the rest of the area to reduce the computation time. The mesh stability verification is shown in S1 Fig.

Reference:

[13] Wang Y, Sah TP, Liu S, Zhai X. Experimental and numerical studies on novel stiffener-enhanced steel-concrete-steel sandwich panels subjected to impact loading. Journal of Building Engineering. 2022;45:103479. doi: 10.1016/j.jobe.2021.103479.

(2) The stabilization time integration step was automatically calculated by ABAQUS/Explicit and can be automatically adjusted during the calculation.

(3) The linear interpolation function was used for the elements of W-shaped steel plate, foam concrete and OSB. Compared with the second-order interpolation function, the displacement solution of LSFC composite slab was more accurate and the calculation time was shorter under large deformation conditions. At the same time, because the W-shaped steel plate was incompressible material, using the linear reduction integral element could avoid the volume self-locking phenomenon to a large extent in elastic-plastic analysis. In the process of finite element simulation, we found that the mesh of this model had distortion. Linear interpolation and reduced integration were used to ensure the accuracy of the calculation.

3.Question:

Describing the FE model in much more detail to ensure its reproducibility, primarily if you will not provide the ABAQUS code or files as supplementary material. For example, what does the sentence mean in line 235 “The bond between the W-shaped steel plate and the foam concrete is assumed to be good” in terms of the FE boundary conditions and connection between meshes? Continuous mesh, bonded contact?

Response: 

(1) We are grateful for this suggestion. To reflect the reviewers' concerns more clearly, we have added more detailed explanations about the FE model. The revised conclusions are as follows:

1) (Lines 418-429)- We further introduce the principal constitutive model for OSB and the five-stage elastoplastic principal constitutive model for steel. 

2) (Lines 442-444)- The flange plate of the W-shaped steel plate is the embedded region, and the foam concrete is the host region. Embedding the flange plate of the W-shaped steel plate in the foam concrete.

3) (Line 465)- Detailed properties of cohesive behavior are added in Table 6.

(2) (Lines 439-441)- [The bond between the W-shaped steel plate and the foam concrete is assumed to be good] this sentence means to define Tie contact boundary conditions on the contact surface of the W-shaped steel plate and the foam concrete, which will glue the meshes of the two surfaces together so that they become a whole and can share the same motion state. In order to express the meaning of the sentence more clearly, we have changed [The bond between the W-shaped steel plate and the foam concrete is assumed to be good] to [Considering that there was no significant separation between the W-shaped steel plate and the foam concrete during the impact process, the contact command between the W-shaped steel plate and the foam concrete was set to tie in order to simplify the model].

4. Question:

Rewriting all the conclusions by adding the numerical values that support them.

Response: 

(Lines 587-608)- Special thanks for your comment. We have rewritten all the conclusions. The revised conclusions are as follows:

(1) Under impact loading, the damage mode of LSFC composite slabs consists of overall bending and local indentation. The impact process is divided into three phases: inertial, loading, and unloading stages.

(2) The maximum deformation of the LSFC composite slab can be reduced by increasing the density and thickness of the foam concrete. When the thickness of foam concrete was increased from 20 mm to 40 mm, the maximum displacement of the LSFC composite slab was decreased by 8.89%. When the density of foam concrete was increased from 450 kg/m3 to 750 kg/m3, the maximum displacement of the LSFC composite slab was decreased by 19.27%. 

(3) Increasing the thickness of foam concrete can increase the energy absorption rate of LSFC composite slabs. However, the energy absorption rate of LSFC composite slabs peaked when the foam concrete density was 600 kg/m3. This indicates that there is an energy absorption critical value for LSFC composite slabs as the density of foam concrete increases. When the density of foam concrete reaches this critical value, the energy absorption rate starts to decrease.

(4) The maximum difference between the impact force and displacement history curves obtained by the FE model and the test results is not more than 13%, which is in good agreement. Meanwhile, the distribution characteristics of stress and strain of W-shaped steel plates are revealed.

(5) The ESDOF model was set up, and the maximum difference in the predicted displacement response of the LSFC composite slab is not more than 9%, which proves that the prediction results of the ESDOF model are reasonable.

5. Question: 

Can you please enlarge the font size? It is really hard to read font size 8 when printed.

Response: 

We apologize for the incorrect use of font size, we have changed the manuscript from 8 to 12 font size.

6. Question:

Line 51-52. "It can not only [...] comprehensive benefit". It is difficult to understand. Please clarify what you mean.

Response: 

(Lines 92-96)- We are sorry for not explaining the meaning of this sentence clearly. The LSFC composite slabs can not only promote the development of green buildings but also have a strong comprehensive benefit due to their unique structure and material combination. The detailed explanations are as follows:

(1) Environmentally sustainable: The use of foam concrete reduces the amount of conventional concrete used, reducing resource consumption and environmental pollution.

(2) Energy efficiency: The use of LSFC composite slabs reduces the self-weight of the building, which in turn reduces the need for infrastructure in the building. In addition, foam concrete has good thermal insulation properties, which can effectively reduce building energy consumption.

(3) Fast construction: The precast board reduces the construction time and labor cost, and improves the construction efficiency.

To make [It can not only promote the development of green buildings but also has a strong comprehensive benefit] easier to understand, we change this sentence to [It not only reduces the use of traditional concrete and resource consumption, but also has good energy-saving efficiency and construction speed, provides a guarantee for the safety and sustainability of the building, and has significant advantages in promoting the development of green buildings, bringing strong comprehensive benefits]. 

7. Question:

Line 54. The word “reasonableness” does not seem common in this context. Do you mean “reliability”?

Response:

(Line 101)- Thank you for your precious comment. The word "reasonableness" means that the ESDOF model has high accuracy in predicting the maximum displacement in the mid-span of LSFC composite slabs. We agree that the word "reliability" can better express this meaning. Therefore, we modify “reasonableness” to “reliability”.

8. Question:

Line 59. Please explain what is the “grip force”. Where is that force acting on?

Response: 

Thanks for your valuable comments. The “grip force” is explained as follows

(1) The grip force refers to the bonding force on the contact surface between the W-shaped steel plate and the foamed concrete.

(2) The grip force acts on the contact surface of W-shaped steel plate and foam concrete.

9. Question:

Section 2.1 Lines 58-65. All parameters included in Table 1 do not need to be described in such detail in the text.

Response:

(Lines 115-117)- Thank you for your helpful comment. The detailed parameters given in Table 1 have been deleted from the text. [All specimens were of the same length and width (length and width of 3000 mm and 450 mm, respectively). The thickness of the W-shaped steel plate is 2 mm, the thickness of the OSB is 9 mm, and the thickness of foam concrete on the top plate is 20 mm, 30 mm and 40 mm. The design values of foam concrete density are 450 kg/m3, 600 kg/m3 and 750 kg/m3, and the measured density values are shown in Table 1] has been changed to [All specimens were of the same length and width. The thickness of the W-shaped steel plate is 2 mm, and the thickness of the OSB is 9 mm].

10. Question:

Section 2.1 Table 1. If possible, please group the rows with the same value for clarity to avoid repeating numbers, especially when only one value, such as the “Impact position” or the hammer mass.

Response:

We think this is an excellent suggestion. Based on your suggestions, we have grouped the rows of Table 1 with the same value.

11. Question: 

(1) Section 2.1 Table 1. Please explain the meaning of the specimen codes in the text.

(2) If the first number beside the letter H, is the thickness of the foam concrete, why not use the whole number in mm, L-H40-#1?

(3) What is “L” standing for?

(4) Adding # to the last number would clearly indicate that it is just numbering the specimens, but you should explain that you are numbering the specimens with shared foam thickness, is this right?

Response: 

Thank you very much for your advice, we fully understand your doubts about this part of the specimen codes and have made explanations and modifications, the details are as follows:

(1) “L” stands for LSFC composite floor slabs. “H” stands for the thickness of the foam concrete. The last number is the ordering for specimens with the same thickness of foam concrete.

(2) I'm sorry that our specimen codes are not clear. We have used the whole number in mm for all specimens.

(3) “L” stands for LSFC composite floor slabs.

(4) We have added # to the last number. As you have considered, when the specimens have the same thickness of foam concrete, the last numbers are the classifications of the specimens.

12. Question: 

Lines 105, 109. The passive impersonal voice is commonly used in technical papers, not “we plotted” or “we can assume”.

Response: 

(Lines 208-209, 215-216)- Thank you for your suggestion. We have changed [We plotted bar graphs showing the maximum value of the slip values at both ends in each specimen, as shown in Fig 6b] to [Bar graphs have been drawn showing the maximum value of the slip values at both ends in each specimen, as shown in Fig 6b].

We have changed [Therefore, we can assume that the end slip is caused by the fragmentation of foam concrete] to [Thus, it was further revealed that the fragmentation of the foam concrete caused the end slip].

13. Question: 

Line 139. Reference [30] does not refer to the D'Alembert principle but to an article describing the inertia effects. Therefore, please move [30] to the end of the sentence.

Response: 

(Line 259)- We were sorry for our mistakes. Thanks for your careful reminder. We've moved the reference to the end of the sentence.

14. Question: 

Line 156. Please rewrite the sentence including “...resistance will be stronger,...”. The whole sentence looks a bit redundant.

Response: 

(Lines 282-285)- Thanks for your suggestion. We have changed [This is because the greater the density of foam concrete, the greater its stiffness and resistance will be stronger, and it is less easy to be compressed and deformed under the impact force so that the buffer performance of LSFC composite slab is weakened] to[This is because as the density of the foam concrete increased, its internal structure became more compact and there was more resistance to the transmission of the impact force, and therefore a greater impact force was generated].

15. Question: 

Formulae (2) and (3). Maybe it is not necessary to show the formula for the mean or specific values.

Response: 

(Lines 300, 431)- As you have considered, the existence of formulae (2) and (3) is indeed redundant and we have removed them.

16. Question:

Line 195. Remove the words “to the outside world”, maybe it is not necessary.

Response: 

(Lines 369)- Thanks for your suggestion. The words “to the outside world” have been removed.

17. Question:

Sections 4.1. and 5.1. “Establishment” is not appropriate to this context. Better use “Setup”.

Response: 

(Lines 404, 521)- Thanks for your careful checks. We are sorry for our mistake. Based on your suggestion, we have changed “Establishment” to “Setup”.

18. Question:

Section 5. Please, write every single word for ESDOF the first time you write it; that is in line 265, not in line 272, where you can write ESDOF.

Response: 

(Line 531)- Thank you for your reminder. We were really sorry for our careless mistakes. We have changed the word “equal-single-degree-of-freedom”, which is not the first occurrence, to "ESDOF". The word "ESDOF" first appears in line 79, so in Section 5, we used ESDOF.

We tried our best to improve the manuscript and made some changes marked in the revised manuscript. We appreciate your work earnestly and hope our corrections will meet with approval. Once again, thank you very much for your comments and suggestions.

Sincerely.

Reviewer #2:

1. Question: 

The exploration and analysis of the experimental data are not sufficiently profound, which does not align with the effort invested in the experiments.

Response: 

We feel great thanks for your professional comment on our manuscript. According to your suggestion, we have made corrections to the manuscript, the detailed corrections are summarized below:

(1) (Lines 175-205)- We detailed the damage mode of LSFC composite slabs, presented the distribution characteristics of foam concrete cracks, and further investigated the extrusion deformation of W-shaped steel plates and the damage at the supports, which are labeled in detail in Fig 5.

(2) (Lines 315-335)- In terms of impact force response, we supplemented the history curves of support reaction force of LSFC composite slabs, as shown in Fig 11. The overall trend of the impact force and support reaction force under impact loading is compared and analyzed from two aspects: response speed and fluctuation, which reveals the force response process.

(3) (Lines 348-358)- In terms of displacement response, we further analyzed the slope and fluctuation of the center displacement history curves of the LSFC composite slabs.

(4) (Lines 500--508)- We also supplemented the stress and strain diagrams of W-shaped steel plates to further reflect the stress characteristics and damage to the W-shaped steel plates, as shown in Fig 19.

2. Question:

The numerical model used for the analysis is relatively simplistic and does not reference state-of-the-art methods.

Response: 

We are very grateful for your constructive suggestion, which has helped us a lot in our finite element simulation. We have tried our best to improve the FE model. At the same time, more details of the FE model have been added. The reasonableness of the FE model is further reflected. The specific modifications are as follows:

(1) (Lines 421-429)- The intrinsic model of low-carbon steel is changed to the five-stage elastoplastic constitutive model. The maximum error of the center peak displacement obtained after the modification is less than 5%;

(2) (Lines 442-444)- Since the flange plate of the W-shaped steel plate was poured into the foam concrete, the flange plate of the W-shaped steel plate is the embedded region, and the foam concrete is the host region. Embedding the flange plate of the W-shaped steel plate in the foam concrete.

(3) (Line 465)- The properties of cohesion behavior were added, as shown in Table 6.

3. Question:

The introduction of the experimental preparation is not comprehensive enough, and the corresponding figures lack detail.

Response: 

Thank you for your valuable suggestion. According to your suggestion, the detailed corrections are summarized below:

(1) (Lines 119-125)- We have added details about the materials and preparation process of the specimens. The details are as follows：

The mixing proportions of cement, water and blowing agent were obtained from tests, as shown in Table 2. The cement type used in the test was P·O42.5 and the blowing agent was FB-602 plant-based cement blowing agent. When preparing the specimens, the molds were first supported around the W-shaped steel plates, the pouring heights were measured and marked inside the molds, and then the prepared foam concrete was poured into the molds. After 28 days of curing, the surface of the foam concrete was evenly coated with nail-free adhesive and the OSB was laid. The specimens can be tested 24 hours later.

(2) (Line 142)- We have added the mixing proportions of foam concrete, as shown in Table 2. 

(3) (Line 172)- The layout of displacement sensors has been added, as shown in Fig 4.

4. Question:

The captions are not clear enough, especially in the inset figures, where the internal scales are entirely illegible, as in Figure 10.

Response: 

We apologize for the lack of clarity in the figures. We have significantly improved the clarity of all the figures to clarify some of the details that were not clear enough in the original.

Thank you for your careful review. We really appreciate your efforts in reviewing our manuscript. We sincerely hope that our modification can meet your expectations.

Sincerely.

Reviewer #3:

1. Question:

In the abstract, summarize the key findings and conclusions in a concise manner.

Response: 

(Lines 14-20)- Thanks a lot for your valuable comment. Following your comment, we have succinctly summarized the key findings and conclusions in the abstract. The specific corrections are as follows：

The tests revealed the failure mode, impact force and displacement response of LSFC composite slabs. The effects of density and thickness of foam concrete and drop height on the peak impact force and energy absorption ratio were investigated. A finite element (FE) model was set up to predict the impact resistance of the LSFC composite slabs, and a good agreement between simulation and test results was achieved. In addition, an equivalent-single-degree-of-freedom (ESDOF) model was set up to predict the displacement response of the LSFC composite slabs under impact loading.

2. Question:

In the introduction, provide more background on the limitations of traditional steel-concrete composite slabs in high-rise buildings to motivate the need for the lightweight alternative studied here.

Response: 

(Lines 35-45)- We sincerely appreciate the valuable suggestion. According to your suggestion, we have modified the introduction section. The specific corrections are as follows：

Currently, the main material for most composite floor slabs is concrete. However, a major problem is the self-weight, long construction period and high cost of concrete[1-3]. For high-rise buildings, due to its length and slenderness ratio is higher than the general buildings, the self-weight is also much larger than the general buildings, resulting in the load transmitted to the foundation is also increased, due to the foundation to withstand the upper part of the larger load, so that the high-rise buildings to resist seismic forces, the wind's ability to greatly reduced. Conventional steel-concrete composite slabs as the main component of the building structure, its self-weight accounted for nearly 40% of the average weight of the entire building structure[4]. If the self-weight of the slabs is reduced, it will be effective in controlling the self-weight of the entire building structure.

Reference：

[1] Jaini ZM, M. Rum RH, Seyed Hakim SJ, Mokhatar SN. Application of Foamed Concrete and Cold-Formed Steel Decking as Lightweight Composite Slabs: Experimental Study On Structural Behaviour. International Journal of Integrated Engineering. 2023;15(2):91-103.

[2] Dawood ET, Hamad AJ. Toughness behaviour of high-performance lightweight foamed concrete reinforced with hybrid fibres. Structural Concrete. 2015;16(4):496-507. doi: 10.1002/suco.201400087.

[3] Shi X, Ning B, Liu J, Wei Z. Effects of re-dispersible latex powder-basalt fibers on the properties and pore structure of lightweight foamed concrete. Journal of Building Engineering. 2023;75:106984. doi: 10.1016/j.jobe.2023.106984.

[4] Luu T, Bortolotti E, Parmentier B, Kestemont X, Briot M, Grass JC, editors. Experimental investigation of lightweight composite deck slabs. 9th International Conference on Steel-Concrete Composite and Hybrid Structures; 2009 Jul 08-10; Leeds, ENGLAND2009.

3. Question:

In Section 2.1, provide a schematic cross-section of the slab with dimensions labeled to help visualize the geometry.

Response: 

(Line 135)- We apologize for not labeling the schematic cross-section of the slab clearly due to our negligence. We have relabeled the schematic cross-section of the slab in Fig 1b.

4. Question:

Specify how the support conditions simulate a simply supported slab.

Response: 

Thanks for your valuable comments. The LSFC composite slabs were directly resting on the support beams during the tests and were not fixed with the support beams. The support beams only provide displacement constraints but not corner constraints. Thus, the simply supported boundary conditions were satisfied.

5. Question:

In Section 3.1, comment on whether the observed failure modes match expectations and any prior work.

Response: 

We sincerely appreciate the valuable comments. The combined failure modes and debonding of LSFC composite slabs were consistent with the predicted results from our previous literature reading and finite element simulation. At the same time, since no shear connector was used in the test, we also predicted that the ends of the LSFC composite slabs would slip. Therefore, the failure modes we observed in our tests were consistent with expectations and any prior work.

6. Question:

Explain the physical meaning/significance of the dip angle change rate parameter.

Response: 

Thanks for your comments. The unique cross-section characteristic of W-shaped steel plates can withstand loading more effectively. The analysis of the dip angle change rate can not only show the transverse deformation characteristic of W-shaped steel plates in the mid-span cross-section when subjected to impact loading, but also show the extrusion deformation of W-shaped steel plates in the mid-span cross-section. This provides a research basis for the optimization of W-shaped steel plates.

7. Question:

Explain the cause of the small fluctuations seen in the loading portion of the force histories 

Response: 

We are very grateful to the reviewer for pointing out the details of the manuscript. The vibration of the slab and the fragmentation of the foam concrete at the impact site caused a change in the contact area between the hammer and the slab, resulting in small fluctuations in the force.

8. Question:

In Section 3.4, discuss how the displacement results compare with any prior studies on similar composite systems.

Response: 

(Lines 348-358)- Thank you for your suggestion. Based on your suggestion, we have added new discussions in section 3.4. The specific corrections are as follows：

The history curves of the center displacement for LSFC composite slabs had a greater slope in the rising stage compared to the conventional SCS composite structures. This is due to the lower overall stiffness of the LSFC composite slab, which allowed it to respond more quickly to external impact force with greater displacement. In the falling stage, the center displacement history curve had less fluctuation, which can be attributed to the fact that the LSFC composite slab had a better energy absorption effect. When the hammer blow acts on the slab surface, the foam concrete was able to absorb most of the impact energy, which made the rebound kinetic energy of the hammer attenuated. At the same time, due to the lightweight nature of the foam concrete, it was able to reduce the overall mass of the slab, which further reduced the inertia of the fluctuations.

9. Question:

Explain why elastic springback causes the final drop in the energy curves.

Response: 

Thank you for your rigorous comment. We have explained your questions as follows:

As the center displacement of the specimen reaches its maximum value, the kinetic energy of the hammer is converted into the internal energy of the specimen, at which time the velocity of the hammer is 0. Subsequently, the hammer begins to rebound, the impact force does negative work, the elastic energy of the specimen begins to be released, and the energy curve is decreased.

10. Question:

In Section 4.1, provide more details on the cohesive element properties used between material layers.

Response: 

(Line 465)- We appreciate your professional advice. We have added detailed properties of cohesive behavior. 

Reference：

[44] Azinović B, Danielsson H, Serrano E, Kramar M. Glued-in rods in cross laminated timber – Numerical simulations and parametric studies. Construction and Building Materials. 2019;212:431-41. doi: 10.1016/j.conbuildmat.2019.03.331.

11. Question:

In Section 4.2, present quantitative validation metrics for the FE model rather than just qualitative statement of good agreement.

Response: 

(Lines 472-484)- Thank you for your constructive comment. The specific corrections are as follows：

Fig 18 gives the vertical permanent deformation of specimen L-H40-#2 obtained from the prediction of the FE model, and the damage characteristics of the test and the FE model were also compared. The results show that the setup FE model can predict the overall and local deformation characteristics of the LSFC composite slabs. The combined deformation pattern of the slabs obtained by the FE model was consistent with the test results. In addition, the FE model can also capture the interlayer separation between OSB and foam concrete. In order to visualize the overall deformation of the slabs, the vertical permanent deformation of the W-shaped steel plate is given in Fig 18b. The peak impact force and peak displacement obtained from the FE model and the tests were compared and are shown in Table 7. It was found that the deviation of the prediction of the peak impact force by the FE model was not more than 13%, and the deviation of the prediction of the peak displacement at the center was not more than 5%. The above analysis proves that the FE model has high accuracy.

12. Question:

Define all variables when the ESDOF model is first introduced.

Response: 

(Lines 522-525)- Thank you for pointing out this omission, we have finished the modification. The specific corrections are as follows：

As shown in 19, the ESDOF model consists of the hammer mass (mh), the effective mass of the LSFC composite slab (ms), the effective mass of the model(me), the impact velocity (v0), the displacement of the model (ut) and the structural stiffness of the model (k).

13. Question:

In Section 5.3.1, explain the physical meaning of the reduction factor used for foam concrete stiffness.

Response: 

Thank you very much for your comment. The foam concrete stiffness reduction factor is a parameter used to adjust the stiffness of foam concrete when considering the non-linear behavior of the material. The value of the stiffness reduction factor is usually between 0 and 1. When the stiffness reduction factor is 1, it means that the stiffness of foam concrete does not undergo any reduction, i.e., the foam concrete exhibits a completely elastic behavior; while when the stiffness reduction factor is close to 0, it means that the stiffness of the foam concrete has been seriously reduced, i.e., the foam concrete exhibits an obvious inelastic behavior. The physical significance of the stiffness reduction factor is to more accurately describe and simulate the deformation behavior of foam concrete under loading.

14. Question:

In Section 5.4, explain the selection of the amplification factor.

Response: 

Thank you for your careful comments. Since the second displacement equation is based on the law of conservation of momentum and the simplified model obtained, while the actual situation exists some factors that do not exactly match the theoretical model. In this experiment, by parameterizing the slope of the displacement history curves obtained from the tests and analyzing the differences with the theoretical curves, the amplification factor “a” was determined after several validations.

15. Question:

In the conclusions, highlight the major findings more clearly.

Response: 

(Lines 587-608)- Thank you for your valuable advice. The modified conclusions are shown below:

(1) Under impact loading, the damage mode of LSFC composite slabs consists of overall bending and local indentation. The impact process is divided into three phases: inertial, loading, and unloading stages.

(2) The maximum deformation of the LSFC composite slab can be reduced by increasing the density and thickness of the foam concrete. When the thickness of foam concrete was increased from 20 mm to 40 mm, the maximum displacement of the LSFC composite slab was decreased by 8.89%. When the density of foam concrete was increased from 450 kg/m3 to 750 kg/m3, the maximum displacement of the LSFC composite slab was decreased by 19.27%. 

(3) Increasing the thickness of foam concrete can increase the energy absorption rate of LSFC composite slabs. However, the energy absorption rate of LSFC composite slabs peaked when the foam concrete density was 600 kg/m3. This indicates that there is an energy absorption critical value for LSFC composite slabs as the density of foam concrete increases. When the density of foam concrete reaches this critical value, the energy absorption rate starts to decrease.

(4) The maximum difference between the impact force and displacement history curves obtained by the FE model and the test results is not more than 13%, which is in good agreement. Meanwhile, the distribution characteristics of stress and strain of W-shaped steel plates are revealed.

(5) The ESDOF model was set up, and the maximum difference in the predicted displacement response of the LSFC composite slab is not more than 9%, which proves that the prediction results of the ESDOF model are reasonable.

16. Question:

Carefully proofread the manuscript to fix minor grammar issues.

Response: 

Thank you for your suggestion. We have tried to do our best to fix the grammar issues. These changes will not influence the content or framework of the manuscript. Instead of listing these changes, we have highlighted them in red in the revised manuscript with track changes. We hope that our revisions will meet with your approval.

17. Question:

Add a table summarizing all specimen details.

Response: 

(Line 137)- Thanks for your suggestion. We have summarized all specimen details in Table 1.

18. Question:

Provide more discussion of how the findings can be utilized for design purposes.

Response: 

(Lines 103-108)- We sincerely appreciate your comment. We have provided further discussion. The detailed discussion follows:

By conducting drop-weight impact tests on LSFC composite slabs. Study the impact energy, foam concrete thickness and density on the dynamic response of LSFC composite slab. Provide design parameters and theoretical basis for the construction of high-rise buildings. Then through the theoretical analysis of the method, accurately meet the slab impact resistance index. To ensure the stability and safety of the building structure under various impact loading, and to improve the survivability of the building in catastrophic events.

We appreciate the enthusiastic work of the reviewer and hope that this correction will be approved. Thank you again for your comments and suggestions!

Sincerely.

---

## [Decision Letter · Decision Letter 1]

10 Dec 2023

Impact response of lightweight steel foam concrete composite slabs: Experimental, numerical and analytical studies

PONE-D-23-29569R1

Dear Dr. Xu,

We’re pleased to inform you that your manuscript has been judged scientifically suitable for publication and will be formally accepted for publication once it meets all outstanding technical requirements.

Kind regards,

Shaker Qaidi

Academic Editor

PLOS ONE

Additional Editor Comments (optional):

Dear Author,

I am pleased to inform you that your manuscript has been accepted for publication in our journal.

The reviewers acknowledged the importance of your work and found that it makes a significant contribution to the field. Your research methods were sound, the data supports the conclusions, and the paper is well-written overall.

Reviewers' comments:

Reviewer's Responses to Questions

**Comments to the Author**

1. If the authors have adequately addressed your comments raised in a previous round of review and you feel that this manuscript is now acceptable for publication, you may indicate that here to bypass the “Comments to the Author” section, enter your conflict of interest statement in the “Confidential to Editor” section, and submit your "Accept" recommendation.

Reviewer #1: All comments have been addressed

Reviewer #3: All comments have been addressed

2. Is the manuscript technically sound, and do the data support the conclusions?

Reviewer #1: Yes

Reviewer #3: Yes

3. Has the statistical analysis been performed appropriately and rigorously? 

Reviewer #1: Yes

Reviewer #3: Yes

4. Have the authors made all data underlying the findings in their manuscript fully available?

Reviewer #1: Yes

Reviewer #3: Yes

5. Is the manuscript presented in an intelligible fashion and written in standard English?

Reviewer #1: Yes

Reviewer #3: Yes

6. Review Comments to the Author

Reviewer #1: (No Response)

Reviewer #3: The authors have investigated substantial changes in the revised manuscript. I believe the manuscript in its current form is suitable for publication.

7. PLOS authors have the option to publish the peer review history of their article (what does this mean?). If published, this will include your full peer review and any attached files.

Reviewer #1: No

Reviewer #3: **Yes: **Mahmoud Akeed

---

## [Editor Report · Acceptance letter]

4 Jan 2024

PONE-D-23-29569R1 

PLOS ONE

Dear Dr. Xu, 

I'm pleased to inform you that your manuscript has been deemed suitable for publication in PLOS ONE. Congratulations! Your manuscript is now being handed over to our production team.

Kind regards, 

on behalf of

Dr. Shaker Qaidi 

Academic Editor

PLOS ONE